# A system-level model for the microbial regulatory genome

Aaron N Brooks[1,2,†], David J Reiss[1,*,†], Antoine Allard[3], Wei-Ju Wu[1], Diego M Salvanha[1,4], Christopher L Plaisier[1], Sriram Chandrasekaran[1], Min Pan[1], Amardeep Kaur[1] & Nitin S Baliga[1,2,5,6,**]

## Abstract

Microbes can tailor transcriptional responses to diverse environmental challenges despite having streamlined genomes and a limited number of regulators. Here, we present data-driven models that capture the dynamic interplay of the environment and genome-encoded regulatory programs of two types of prokaryotes: *Escherichia coli* (a bacterium) and *Halobacterium salinarum* (an archaeon). The models reveal how the genome-wide distributions of *cis*-acting gene regulatory elements and the conditional influences of transcription factors at each of those elements encode programs for eliciting a wide array of environment-specific responses. We demonstrate how these programs partition transcriptional regulation of genes within regulons and operons to re-organize gene–gene functional associations in each environment. The models capture fitness-relevant co-regulation by different transcriptional control mechanisms acting across the entire genome, to define a generalized, system-level organizing principle for prokaryotic gene regulatory networks that goes well beyond existing paradigms of gene regulation. An online resource (http://egrin2.systemsbiology.net) has been developed to facilitate multi-scale exploration of conditional gene regulation in the two prokaryotes.

**Keywords** EGRIN; gene regulatory networks; systems biology; transcriptional regulation

**Subject Categories** Genome-Scale & Integrative Biology; Transcription; Computational Biology

**Mol Syst Biol. (2014) 10: 740**

## Introduction

Deciphering how microbes colonize dynamically changing environmental niches with few regulators and streamlined genomes will require mechanistic and system-level characterization of their gene regulatory networks (GRNs). Even a streamlined microbial genome encodes an intricate network of regulatory and signaling systems that sense and process extracellular and intracellular information to regulate gene expression at multiple levels (transcriptional, post-transcriptional, translational, allosteric, etc.). A significant fraction of these environmental signals are relayed by transcription factors (TFs) that modulate transcriptional activity when they bind DNA. TFs typically bind conserved, ~6–20 nucleotide DNA sequences located in intergenic regions immediately adjacent to transcription initiation sites. These TF-binding sites are referred to as gene regulatory elements (GREs).

A goal of systems biology has been to map the complete set of TFs, GREs, and their interactions, using high-throughput techniques including ChIP-chip (Blat & Kleckner, 1999), yeast two-hybrid (Fields & Song, 1989), DNase I hypersensitivity (Crawford *et al*, 2004), or more modern variants using sequencing (Johnson *et al*, 2007). In parallel, attempts have been made to infer GRNs directly from gene expression data (Segal *et al*, 2003; Bonneau *et al*, 2007; Faith *et al*, 2007; De Smet & Marchal, 2010). Such high-throughput approaches are attractive because they would accelerate discovery in understudied organisms by circumventing significant labor and cost.

Inference of system-scale GRNs that are both predictive and mechanistically accurate, however, has proven difficult for a number of reasons, including: (1) the statistical challenge of confidently discovering GREs across the genome, *de novo*; (2) the consequences of non-linear gene regulatory dynamics, including combinatorial molecular interactions at gene promoters; and (3) the often non-canonical locations of GREs throughout the genome (including internal to operons and within coding sequences). A remaining challenge, therefore, is to produce an unbiased map of

1 Institute for Systems Biology, Seattle, WA, USA
2 Molecular and Cellular Biology Program, University of Washington, Seattle, WA, USA
3 Département de Physique, de Génie Physique et d'Optique, Université Laval, Québec, QC, Canada
4 LabPIB, Department of Computing and Mathematics FFCLRP-USP, University of Sao Paulo, Ribeirao Preto, Brazil
5 Departments of Microbiology and Biology, University of Washington, Seattle, WA, USA
6 Lawrence Berkeley National Laboratories, Berkeley, CA, USA
 *Corresponding author. Tel: +1 206 732 1391; Fax: +1 206 732 1299; E-mail: dreiss@systemsbiology.org
 **Corresponding author. Tel: +1 206 732 1266; Fax: +1 206 732 1299; E-mail: nbaliga@systemsbiology.org
 †These authors contributed equally to this work

TF-binding site locations throughout the genome, including information about what binds to those sequences, in what contexts they are bound, and, importantly, how TF-binding throughout the genome ultimately influences cellular physiology.

We previously constructed an "Environment and Gene Regulatory Influence Network" (EGRIN) for *Halobacterium salinarum NRC-1* (Bonneau *et al*, 2007). This model was constructed in two steps. First, modular organization of gene regulation was deciphered through semi-supervised biclustering of gene expression, guided by biologically informative priors and *de novo cis*-regulatory GRE detection for module assignment (cMonkey; Reiss *et al*, 2006). Second, using a regression-based approach, transcriptional changes of genes within each bicluster were modeled as a linear combination of influences of TFs and environmental factors (Inferelator; Bonneau *et al*, 2006). While full description of these algorithms is beyond the scope of this work, readers are encouraged to refer to the original papers and Supplementary Information for more detail.

The EGRIN networks learned by cMonkey and Inferelator accurately predicted transcriptional changes in new environments, a feat that has subsequently been replicated by other network inference strategies (Faith *et al*, 2007; Lemmens *et al*, 2009; Marbach *et al*, 2012); yet, these network models have failed to capture detailed regulatory mechanisms that operate only in specific environments, at non-canonical genomic locations, or in complex combinatorial schemes.

Here, we report significant advancement to inference of GRNs that overcomes many of these challenges. We have developed a methodology applicable to any sequenced microbe in culture to infer EGRIN 2.0 models for two representative organisms from the primary branches of prokaryotic life—bacteria and archaea: (1) *Escherichia coli*, a bacterium with a wealth of information about transcriptional regulatory mechanisms and related experimental data (Salgado *et al*, 2012); and (2) *H. salinarum*, an archaeon with few examples of regulatory mechanisms that have been characterized in detail, but extensive experimental data from recently conducted systems biology studies (Bonneau *et al*, 2007; Koide *et al*, 2009). The wide range of prior knowledge for these organisms proved invaluable for testing our model. In addition, we have also conducted new experiments that validate EGRIN 2.0-predicted complex modulation of the *E. coli* transcriptome structure during varying stages of growth in rich media.

EGRIN 2.0 models the organization of GREs within every promoter and their distributions across the entire genome—even in non-canonical locations—and links the contexts in which they act to conditional co-regulation of genes. These features are formalized in EGRIN 2.0 by condition-specific, co-regulated modules or corems. Corems are overlapping sets of co-regulated genes that, in some cases, group together genes from different regulons and, in other cases, subdivide genes of the same regulon, or even the same operon. EGRIN 2.0 formalizes how the genome-wide coordination of previously characterized and newly discovered regulatory mechanisms dynamically associates genes into corems, bringing together functionally related genes from different operons and regulons whose deletions have similar impact on cellular fitness. Our results show how prokaryotes, much like eukaryotes, can produce complex gene expression patterns with a relatively small number of regulatory components.

# Results

## Construction of EGRIN 2.0 models

We developed an ensemble framework that models the condition-specific global transcriptional state of the cell as a function of combinations of transient TF-based control mechanisms acting at intergenic and intragenic promoters across the entire genome. Specifically, for each of the two organisms, *H. salinarum* and *E. coli*, we aggregated associations across genes, GREs, and environments from many individual EGRIN models, each trained on a subset of the gene expression data, to: (1) quantify confidence in each model-predicted association; (2) reveal context-dependent regulatory mechanisms that occur infrequently in the data; and (3) discover non-canonical regulatory mechanisms. We refer to the aggregated, post-processed ensemble of EGRIN models as EGRIN 2.0 and conditionally co-regulated modules as corems (details provided in Materials and Methods, Fig 1; ensemble statistics available in Supplementary Table S3). For *E. coli*, we generated two models: one trained on an expression compendium from Lemmens *et al* (2009) and the other trained on a dataset from the DREAM5 consortium (Marbach *et al*, 2012). We used the model trained on DREAM5 data to compare model performance (described below).

## EGRIN 2.0 discovers experimentally characterized regulatory mechanisms

A high-quality GRN has to be both comprehensive (high recall) and accurate (high precision). To evaluate the quality of EGRIN 2.0, we compared its predictions on *E. coli* to RegulonDB (Gama-Castro *et al*, 2011), an extensive, manually curated, gold-standard of experimentally validated TF–gene interactions. For our comparison, we used a version of RegulonDB curated by the DREAM5 consortium. We compared the genome-wide distribution of each *de novo* discovered GRE in EGRIN 2.0 (trained on DREAM5 data expression compendium) to experimentally characterized binding locations of every TF in RegulonDB. This comparison showed that EGRIN 2.0 had accurately located binding sites for 60% of experimentally characterized TFs in RegulonDB (53 out of 88 at FDR ≤ 0.05 for all TFs with ≥ 3 unique sites; see Materials and Methods). At a standard precision cutoff of 25%, EGRIN 2.0 recovered 555 "strong evidence" TF–gene interactions, which is 2.7X as many validated interactions as algorithms that exclusively use expression data, that is, without genomic sequence information (Fig 2A, Supplementary Figs S8, S9 and S10, Supplementary Dataset S3, Materials and Methods; Faith *et al*, 2007; Marbach *et al*, 2012). As expected, the ensemble network had greater precision and recall than individual *cMonkey* runs. Furthermore, integration of *Inferelator*-predicted TF influences with GRE-based predictions increased overall algorithm performance. The increased performance observed in the integrated model may be due to its ability to detect regulatory events that do not depend on a linear relationship between TF expression and target gene expression (which is assumed for most "direct" methods, like those in the DREAM5 ensemble network). These results show that integrating complementary methods, such as regression-based inference of TF regulation, biclustering-based inference of network modularity, and *de novo* GRE detection, improve the accuracy and coverage of the inferred GRN.

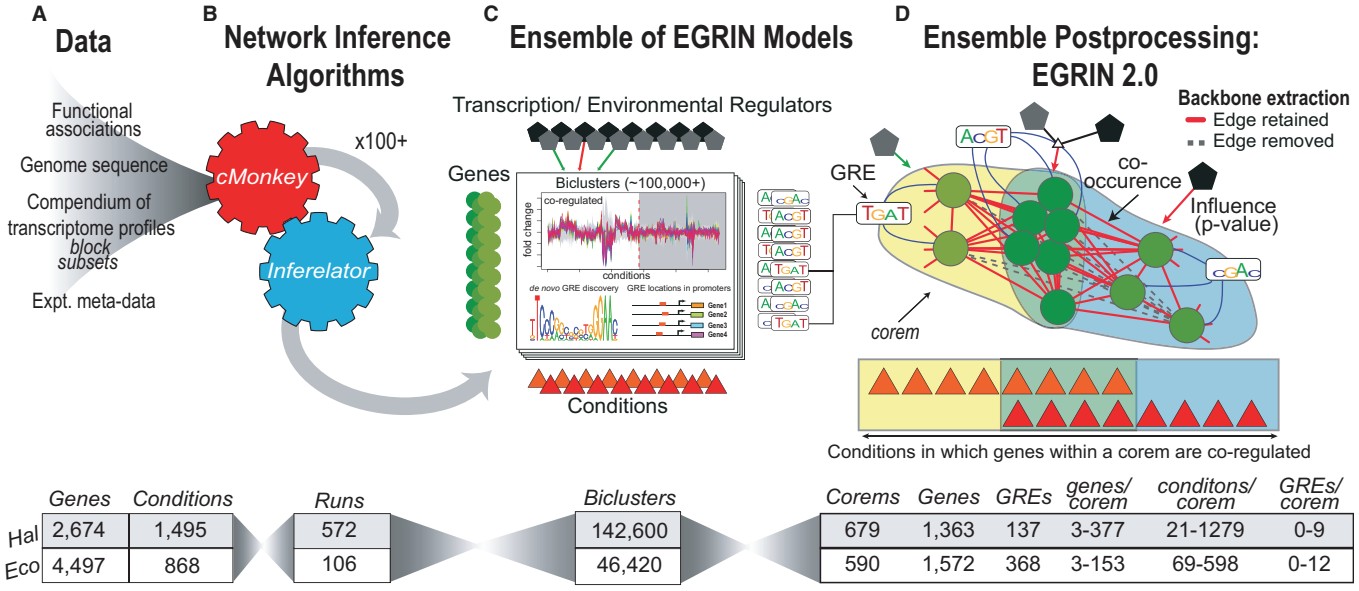

| | Genes | Conditions | | Runs | | Biclusters | | Corems | Genes | GREs | genes/ corem | conditons/ corem | GREs/ corem |
|---|---|---|---|---|---|---|---|---|---|---|---|---|---|
| Hal | 2,674 | 1,495 | | 572 | | 142,600 | | 679 | 1,363 | 137 | 3-377 | 21-1279 | 0-9 |
| Eco | 4,497 | 868 | | 106 | | 46,420 | | 590 | 1,572 | 368 | 3-153 | 69-598 | 0-12 |

**Figure 1.  EGRIN 2.0 model construction.**

Workflow summary for EGRIN 2.0. Tables below each panel contain detailed statistics for the *Halobacterium salinarum* and *Escherichia coli* models. See also Supplementary Fig S1.

A, B   The *cMonkey* and *Inferelator* algorithms were applied many times to subsets of gene expression data from large compendiums of transcriptome profiles to construct many individual *EGRIN* models.

C   Individual *EGRIN* models were integrated into an ensemble for filtering, querying, and ranking relationships among genes (circles), regulators (pentagons), motifs (sequence logos), and the conditions (triangles) in which these relationships were discovered.

D   The library of relationships was mined using algorithms for motif clustering, backbone extraction, and community detection to construct the final EGRIN 2.0 model. In EGRIN 2.0, overlapping co-regulated sets of genes (corems, shaded regions of the graph) are statistically associated with specific gene regulatory elements (GREs, sequence logos, blue edges), regulatory influences (pentagons, green or red depending on direction), and environments in which they are co-regulated (triangles). Each node represents a gene in the model. Genes are connected via co-regulation edges, with weights that reflect the number of occurrences in the ensemble. Dashed edges were removed from the model by backbone extraction.

Since few GREs have been characterized in *H. salinarum*, we performed a global assessment and discovered that GREs in EGRIN 2.0 occur at consistent locations across many gene promoters throughout the genome (Supplementary Fig S3). We could even assign putative roles for some GREs based on their location relative to transcription start sites (TSSs). For instance, the location of TATA box-like elements (GRE #25) between −21 and −40 nucleotides upstream of TSSs in *H. salinarum* is consistent with the characterized location of basal elements in archaeal promoters (TFB/TBP complex recognition sites) (Geiduschek & Ouhammouch, 2005). Similarly, other elements occurred either consistently downstream of the TATA box (putative repressors, e.g., GRE #1 and #2) or upstream of these basal elements (putative activators, e.g., GRE #5). Thus, even in organisms where genome-wide TF-binding data are scarce, EGRIN 2.0 can be used to infer and predict putative roles for *de novo* discovered GREs.

### Corems model genes with similar effects on organismal fitness

We investigated whether the model goes beyond simple co-expression to group together genes that have similar phenotypic contributions. We did this because previous studies have reported weak correlation between gene expression and fitness (Price *et al*, 2013). For all genes in each corem, we computed pairwise correlations of fitness effects in a dataset generated from a survey of relative growth rates for 3,902 single gene deletion strains of *E. coli* subjected to a chemical genomics screen spanning 324 different environmental conditions (Nichols *et al*, 2011). We discovered that more than one-third of gene-pairs with the most similar fitness effects across environments (Pearson correlation > 0.75) were grouped together in corems. We evaluated significance of this result by performing similar analysis using modules based on co-expression (WGCNA; Langfelder & Horvath, 2008) and regulons (RegPrecise and RegulonDB), where a regulon is defined as a set of genes regulated by the same TF. While WGCNA and regulons also grouped significant numbers of high fitness-correlated gene-pairs (one-sided KS-test < 0.05), corems were more enriched for highly similar fitness associations (higher KS D-statistic) and in general provided greater precision and coverage (Fig 2B). As an example, corems group together 5X as many gene-pairs with highly correlated fitness effects as RegPrecise, RegulonDB, or WGCNA (134 out of 185 gene-pairs with Pearson correlation ≥ 0.9 are discovered in corems, Supplementary Dataset S4). Most importantly, corems retained a high degree of enrichment for gene-pairs with highly correlated fitness effects after removing all associations attributable to operon and regulon memberships, and even combinatorial control (Supplementary Fig S14, Supplementary Dataset S4). This suggested that corems model regulatory associations among genes that cannot be explained within the existing paradigms of regulons and operons.

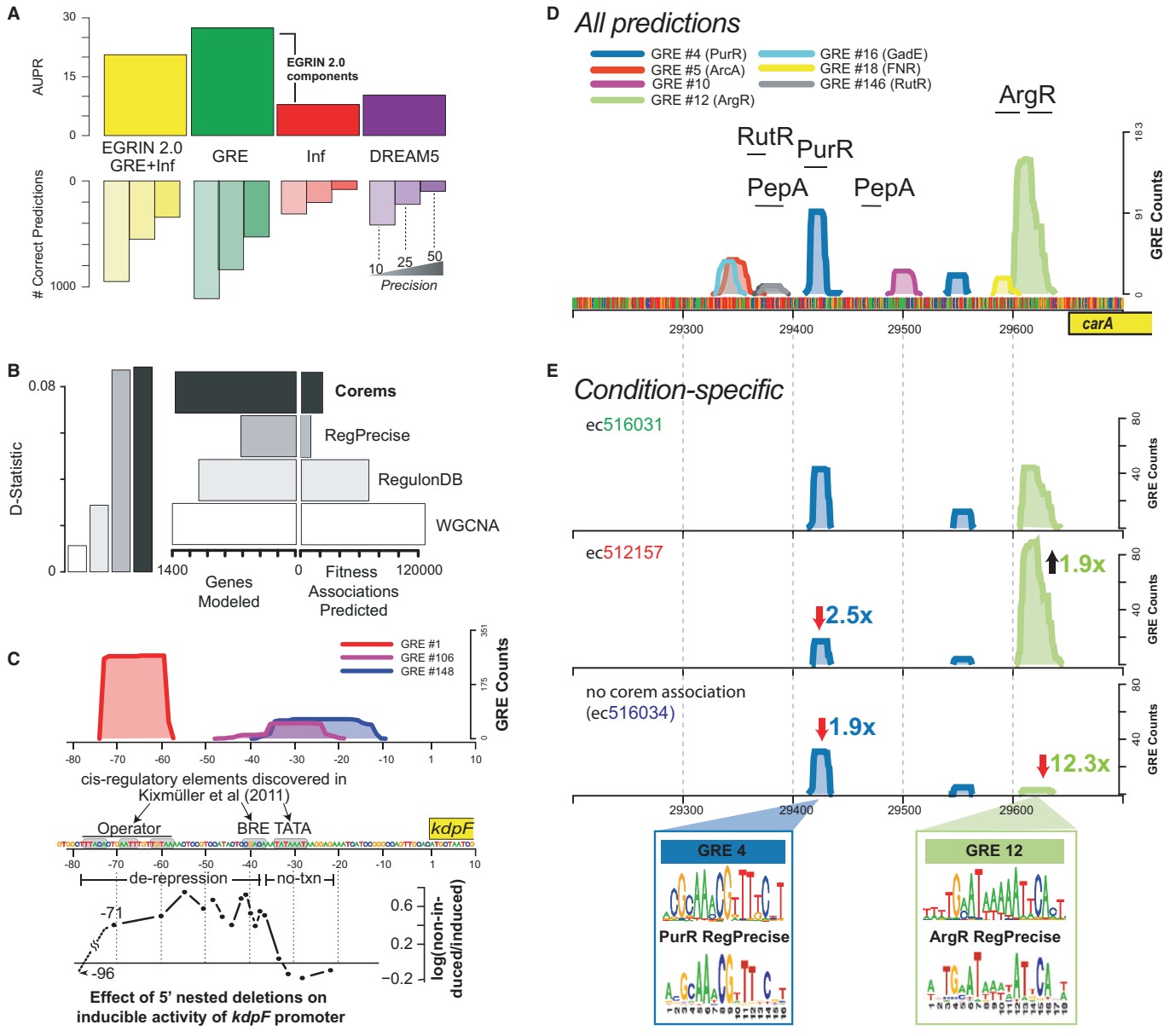

**Figure 2.  EGRIN 2.0 model validation.**

Two global and two specific validations of EGRIN 2.0's ability to infer accurate GRNs.

A  EGRIN 2.0 performance on experimentally validated gold-standard network. Comparison of EGRIN 2.0 model components ("GRE": GRE-only; "Inf": *Inferelator*-only) to CLR and the DREAM5 community ensemble network, against RegulonDB (strong evidence code). (Top) Area under the precision-recall curve (AUPR) and (bottom) number of correct predictions at 10, 25 and 50% precision.

B  Enrichment of similar fitness effects within gene modules. (Left) Magnitude of enrichment for gene-pairs with similar fitness consequences, assessed by one-tailed KS-test (KS D-statistic). (Right) Number of genes and gene-pairs predicted by each method. Comparison methods include EGRIN 2.0 corems, co-expression modules from WGCNA, and regulons from databases (RegPrecise and RegulonDB).

C  Promoter architecture of the *Halobacterium salinarum kdpFABC* promoter predicted by the EGRIN 2.0 model. (Top) Frequency of GRE alignment to each position in the *kdpFABC* promoter. GREs are indicated by shaded lines. (Middle) Genome sequence marked with putative functions by Kixmuller *et al* (2011). (Bottom) Transcriptional activity measurements from truncated promoters used by authors to validate these sites.

D  Predicted architecture of the *Escherichia coli carA* promoter across all ensemble predictions (as in C). Horizontal bars above peaks mark experimentally characterized TF-binding sites (RegulonDB). Significant GRE matches to characterized *E. coli*-binding sites in RegulonDB are indicated in parentheses.

E  Condition-specific states of the *carA* promoter in *E. coli*. Variation in conditional discovery of GREs (counts and fold-change relative to ec516031, top) suggests when they are "active" across three different subsets of experimental conditions in the *carA* promoter. (Bottom) Condition subsets correspond to co-regulation of *carA* with genes in the nucleotide and pyrimidine corems (ec516031, ec512157) or environments where *carA* is not co-regulated with genes in any corem (ec516034). Motif logos for GRE #4 (PurR) and GRE #12 (ArgR) from the EGRIN 2.0 predictions compared to logos from RegPrecise.

In other words, corems group together genes that are regulated by distinct TFs. For example, the ArgR-regulated acetylglutamate kinase, *argB*, and *ilvC*, an IlvY-regulated ketol-acid reductoisomerase, have fitness correlation of 0.95 (Pearson coefficient), which suggests an important coupling between branched-chain amino acid biosynthesis and arginine metabolism (Table 1). Although these genes are regulated by distinct TFs (ArgR and IlvY, respectively), the high similarity of their expression changes across multiple environments brings them together into the same corem (ec*512157*). There are 319 highly correlated (Pearson correlation ≥ 0.75) fitness associations among genes from different regulons that are modeled by corems—each of which suggests an important physiological coupling that results from the coordinated activity of TFs (Supplementary Dataset S5). These examples illustrate how the organizing principle of corems captures fitness-relevant associations within a GRN that are overlooked by current definitions for gene–gene co-regulation, such as regulon and operon.

## EGRIN 2.0 predicts detailed organization and context-specific importance of GREs in gene promoters

We next investigated accuracy of EGRIN 2.0-predicted spatial organization of GREs and their context-specific roles in mediating transcriptional regulation from specific promoters. We did this analysis in context of one of the best studied *H. salinarum* promoters: *kdpFABC*, with data not used for model training. The *kdp* operon encodes an ATP-dependent potassium transporter that counterbalances extremely high salinity in the extracellular environment. EGRIN 2.0 predicts that at least three GREs are putatively responsible for mediating transcriptional regulation of this operon: GRE #1, GRE #148, and GRE #106 (Fig 2C). The locations of these GREs align to regions that were experimentally characterized in an independent study as "Operator" and "BRE-TATA" elements, respectively. This demonstrates that EGRIN 2.0 is able to accurately predict the organization of GREs in gene promoters at nucleotide resolution.

Since these sites also had characterized transcriptional roles [determined by promoter truncation experiments (Kixmuller *et al*, 2011)], we asked whether EGRIN 2.0 would have been able to predict these roles given the context in which the GREs were discovered. Strikingly, we find that GRE #1 (aligned to the "Operator") was discovered in environments, including low salt (hypergeometric

FDR = $6.9 \times 10^{-12}$), where the transcript is repressed (one-sided *t*-test $P = 0.048$), while GRE #106, which aligns to the "BRE-TATA" region, was discovered in environments, including low oxygen (hypergeometric FDR = $1.8 \times 10^{-9}$), where transcript levels are elevated (one-sided *t*-test $P = 1.2 \times 10^{-3}$; Supplementary Information). Here onwards, we will refer to a GRE as "*active*" when it is predicted to be important for transcriptional regulation at a specific promoter (see Supplementary Fig S6 for details). The environmental contexts in which the three GREs in the *kdp* promoter are predicted to be active are especially interesting because perturbations to external potassium levels and energy-producing mechanisms have been shown to significantly influence expression of this operon (Wurtmann *et al*, 2014). Thus, EGRIN 2.0 had accurately predicted that a trade-off in relative influence of GRE #1 (repressing) versus GRE #106 (activating) controls expression levels of this operon in a condition-specific manner, exactly as was characterized by independently performed experiments. This is powerful because it shows that using EGRIN 2.0 we can predict when (context) and how (activate or repress) a specific GRE(s) within a promoter might act, even though we might not know the precise regulatory mechanism (e.g., TF binding/unbinding, allosteric activation, co-factor interaction, etc.).

## Conditionally active GREs within each promoter reorganize gene memberships within corems

We investigated whether EGRIN 2.0 accurately links the same GRE at different promoter locations, the environments in which it is predicted to be active within each of those promoters, and conditional co-regulation of the associated genes (see Supplementary Information). We did this analysis with genes of nucleotide biosynthesis in *E. coli,* including key branch-point enzymes *carA* (b0032) and *pyrL* (b4246), since they are canonical, extremely well-studied pathways that are critical for survival. Regulation of *carA*, which catalyzes synthesis of an important metabolic intermediate in several amino acid and nucleotide metabolism pathways (carbamoyl phosphate), is known to be sensitive to purine and pyrimidine pools, as well as arginine (Neidhart, 1996). EGRIN 2.0 discovered several previously characterized and new mechanisms for regulation of *carA*, including two GREs (GRE #4 and GRE #12) that match to consensus sequence motifs for PurR and ArgR, respectively

**Table 1.   Corems group together genes from different regulons with highly correlated fitness effects**

| Gene 1 | Gene 2 | Fitness correlation | Regulon gene 1 | Regulon gene 2 | Corems |
|--------|--------|---------------------|----------------|----------------|--------|
| *b3774* | *b3959* | 0.959012 | IlvY | ArgR | 512157 |
| *b2913* | *b3829* | 0.938764 | PurR | MetR | 512157 |
| *b3829* | *b3959* | 0.934393 | MetR | ArgR | 512157;554056 |
| *b2913* | *b3941* | 0.932025 | PurR | MetR | 512157 |
| *b3957* | *b3941* | 0.931565 | ArgR | MetR | 512157;554056 |
| *b3172* | *b3829* | 0.930382 | ArgR | MetR | 512157;554056 |
| *b2913* | *b3774* | 0.927776 | PurR | IlvY | 512157;512477 |
| *b3941* | *b3774* | 0.927251 | MetR | IlvY | 512157 |
| *b3960* | *b3941* | 0.921375 | ArgR | MetR | 512157;554056 |
| *b3941* | *b3959* | 0.921282 | MetR | ArgR | 512157;554056 |

     

(Piette *et al*, 1984) (Fig 2D). Remarkably, EGRIN 2.0 discovered novel overlapping organization of GRE #4 and GRE #12 in the *pyrL* promoter that was not previously reported in RegulonDB (Supplementary Fig S18). This promoter organization was verified upon mapping overlapping binding sites for ArgR and PurR precisely at the predicted locations in ChIP-chip data that were not used in model training (Cho *et al*, 2011, 2012).

We were most interested, however, to understand the consequences of conditional regulation at ArgR and PurR-associated GREs on variable expression of *carA* in different environments. Indeed, EGRIN 2.0 predicts three condition-specific states of the *carA* promoter with respect to when PurR- and ArgR-matched GREs are conditionally active: (1) high PurR and high ArgR; (2) low PurR and high ArgR; and (3) high PurR and low ArgR (Fig 2E). Interestingly, two of these promoter states correspond to co-regulation of *carA* with a different combination of genes (i.e., different corems), functionally separating pyrimidine from purine biosynthesis (Fig 4B), while the third state is not associated with co-regulation of *carA* with the genes of any corem. Thus, the context in which GREs are active accurately explains when and how genes are co-regulated in different overlapping combinations to perform distinct functions.

## Conditionally active GREs within operons generate multiple, overlapping, and differentially regulated transcript isoforms

Some of the GREs discovered in EGRIN 2.0 occur in non-canonical locations and lead to unexpected transcriptional behaviors, such as the subdivision of operons into multiple transcriptional units. Previously, we reported pervasive modulation of the *H. salinarum* transcriptome structure by transcriptional elements that are located within operons and coding regions (Koide *et al*, 2009). EGRIN 2.0 recapitulated this phenomenon by sub-dividing operon genes into different corems. In all, the model predicted that nearly one-third of all *H. salinarum* operons generate multiple transcript isoforms (Supplementary Figs S11, S12, and S13, Supplementary Information for details). Nearly half of these predictions of conditional operon structures were corroborated by experimentally mapped transcriptional breaks (hypergeometric $P = 4.2 \times 10^{-3}$; Supplementary Dataset S6; Koide *et al*, 2009). Often, these transcript boundaries were adjacent to GREs that coincide with experimentally determined TFB-binding sites (Facciotti *et al*, 2007; Fig 3A and B), reinforcing the accuracy of EGRIN 2.0 predictions.

We further investigated whether EGRIN 2.0 provides insight into downstream consequences of differentially regulating multiple transcript isoforms from the same operon. The *dppAB1C2-oppD2-ykfD-VNG2342H* operon (hereafter the "*dpp* operon") in *H. salinarum* encodes an ATP-dependent dipeptide transporter. Some periplasmic binding proteins (like *dppA*) have the reported ability to function in conjunction with different ABC transport systems, giving support to the hypothesis that *dppA* can be regulated independently (Higgins *et al*, 1990). Despite high co-expression of the entire operon in the training data (mean $R^2 = 0.6$ across 1,495 conditions), EGRIN 2.0 predicted that the genes of this operon are transcribed as three different isoforms, each co-regulated with genes of a different corem: (1) the entire operon (hc21645—"*dpp* corem"); (2) the entire operon except the leader gene, *dppA* (hc21279—"*permease* corem"); and (3) just *dppA* (hc6326—"*leader corem*"). These predicted isoforms were verified by experimentally mapped transcript boundaries (Fig 3B). Each of these corems contains a different *dpp* isoform and is enriched for a different biological function, including vitamin biosynthesis, porphyrin metabolism, and purine biosynthesis, respectively (Fig 3C). Predicted differential regulation of the core permease (*dppB1C2-oppD2-ykfD-VNG2342H*) with porphyrin metabolism genes in the *permease* corem is consistent with the reported capability of this transporter system to uptake heme when it functions with a *different* solute binding protein (i.e., without dppA; Letoffe *et al*, 2006). Overall, EGRIN 2.0 provided insight into the distinct environment-dependent functional associations of each transcript isoform.

Further, EGRIN 2.0 revealed that segmentation of the *dpp* operon into multiple corems is mediated by conditionally active GREs located both upstream and internal to the operon. For example, EGRIN 2.0 predicted that GRE #6 was responsible for disassociating *dppA* transcription from the remainder of the operon. Interestingly, GRE #6 was also discovered in the promoters of nearly all of the other genes in the *leader* corem (Fig 3, Supplementary Fig S23, Supplementary Dataset S7). Similarly, GRE #1 was implicated in co-regulating the permease-encoding transcript with other genes in the *permease* corem, and GRE #17 for co-regulating the entire operon with other genes in the *dpp* corem. EGRIN 2.0 also predicted specific segmentation pattern of the *dpp* operon during "lag growth phase". This prediction was verified upon observing that a transcript break appears downstream to *dppB1* precisely when a batch culture transitions from lag to log phase of growth (indicated by arrow in Fig 3B heatmap). This is just one of 98 operons with experimentally validated conditional isoforms in *H. salinarum*. For each instance, a similar correspondence between mechanism, context, and function could be demonstrated (Supplementary Figs S19, S20 and S21 and online). Interestingly, even in *E. coli*, where previous studies report a single transcript for the *dpp* operon (Abouhamad & Manson, 1994), EGRIN 2.0 discovered that it is actually transcribed as multiple, condition-specific transcript isoforms, each of which participates in a different physiological process (Supplementary Figs S11, S12 and S13).

While we were aware of extensive transcriptional heterogeneity within operons in *H. salinarum*, we were surprised that EGRIN 2.0 predicted that the same phenomenon also occurred extensively in *E. coli*. To see whether this were true, we mapped the *E. coli* global transcriptome structure across varying phases of growth in rich media using a densely tiled microarray (see Materials and Methods). We used this new gene expression dataset to identify the corems in which different combinations of operon genes (i.e., transcript isoforms) were co-regulated in some or all phases of growth and to characterize transcriptional breaks using previously developed methodologies (Koide *et al*, 2009). We observed transcriptional breaks in nearly 20 percent of operons (including the *E. coli dpp* operon) just over this 9-time point growth study, validating EGRIN 2.0 prediction that nearly one-quarter of all *E. coli* operons have conditional isoforms during varying stages of growth (hypergeometric $P = 1.07 \times 10^{-5}$, Supplementary Figs S11, S12 and S13, Supplementary Dataset S6). Experimental validation of this enormous transcriptional heterogeneity among operons in *E. coli* demonstrates the power of EGRIN 2.0 to distinguish nuanced patterns in complex data and provide both mechanistic explanation and context for when and why the novel phenomena might occur.

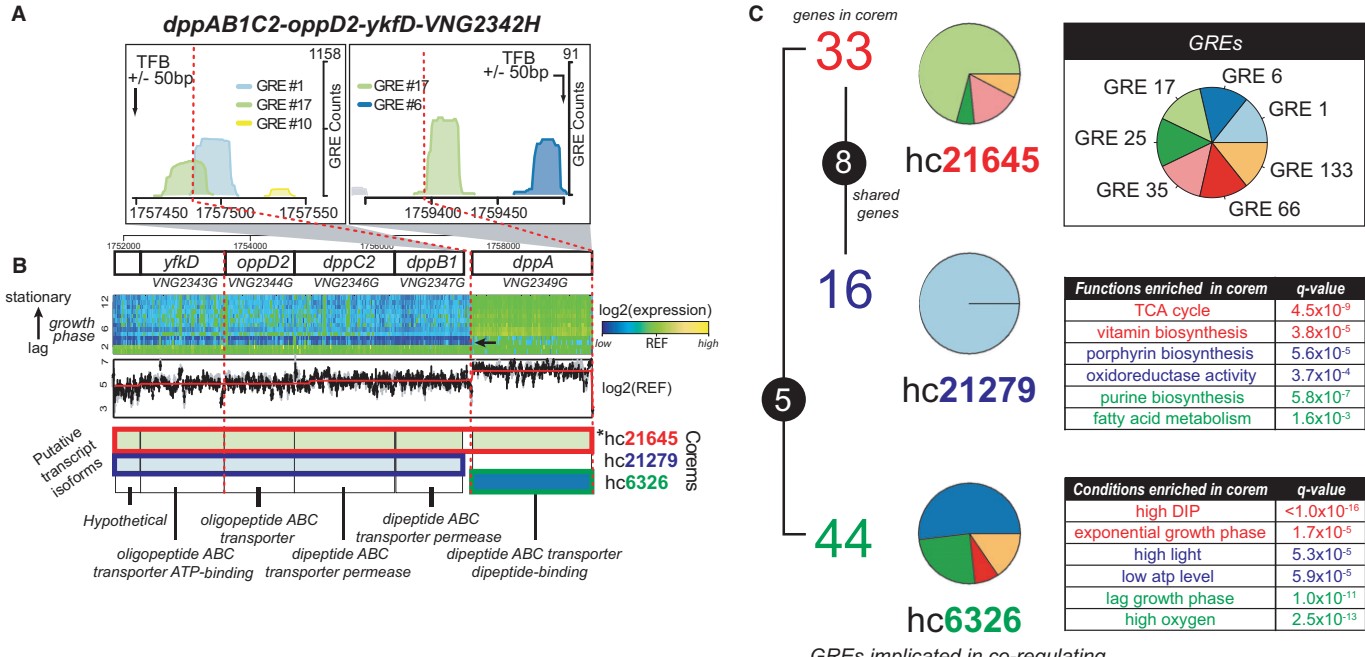

**Figure 3.  Conditional influences at GREs within canonical and non-canonical promoters differentially regulate multiple transcript isoforms from the same operon (*Halobacterium salinarum*).**

Three GREs and three corems explain the condition-specific expression of transcriptional isoforms from an operon in *H. salinarum*. Model predictions are supported by high-resolution tiling array and ChIP-chip data.

A   (Top) Predicted GREs located within (left) and upstream of (right) the *H. salinarum dpp* operon. Locations of experimentally mapped TFB-binding sites (vertical arrows; Facciotti *et al*, 2007) and experimentally mapped transcription break sites (vertical red dashed lines, see B; Koide *et al*, 2009) are indicated. (Bottom) Locations of predicted GREs relative to coding segments of the *dpp* operon.

B   (Top) Expression changes during growth in the genomic region covering the *dpp* operon measured by high-resolution tiling microarray. (Middle) Raw RNA hybridization signal from mid-log growth phase. (Bottom) Three predicted transcripts from the *dpp* operon. Internal colors correspond to the GREs (as in A) putatively responsible for regulating each transcript (derived from corem membership in C). Boxed colors indicate corem membership for each transcript (described in C). Red dashed lines indicate experimentally measured transcription break sites. Transcriptional break at lag phase highlighted by an arrow. Functional annotation for each gene located at bottom.

C   (Left) Three *H. salinarum* corems model differential regulation of *dpp* operon isoforms: (1) the entire operon (hc21645—"*dpp* corem"; top); (2) five tail genes, excluding *dppA* (hc21279—"*permease* corem"; center); and (3) the leader gene, *dppA* (hc6326—"*leader corem*"; bottom). Colored numbers denote quantity of genes in each corem; numbers in black shaded circles indicate the number of genes shared between corems. Pie charts represent average predicted influence of GREs on the regulation of genes in each corem (see Supplementary Fig S6 for detail). (Top-right) Pie chart key indicates GRE identity. (Bottom-right) Tables list enriched gene functions (Dennis *et al*, 2003) and environmental conditions for each of the corems (computed using the environmental ontology; see Materials and Methods and Supplementary Information).

## Some TFs act similarly across certain environments to co-regulate functionally related subsets of genes across their respective regulons

We investigated whether EGRIN 2.0 provides insights into context-dependent differential regulation of branched metabolic pathways—even those that have been meticulously studied for decades, such as *de novo* biosynthesis of nucleotides in *E. coli* (Neidhart, 1996). At least seven GREs were implicated in partitioning (purine biosynthesis: ec516034—"*purine* corem"; pyrimidine biosynthesis: ec512157—"*pyrimidine* corem") or co-regulating (ec516031—"*nucleotide* corem") nucleotide biosynthesis into multiple overlapping corems (Fig 4A, Supplementary Figs S22 and S23). The genome-wide locations for four of these GREs significantly overlapped with known binding locations for PurR, ArgR, MetJ, and IclR. Partitioning and co-regulation of purine and pyrimidine biosynthesis can be attributed to the location of these GREs in promoters of pathway genes,

including *carA*, and the environments in which they are predicted to be active (Fig 2D and E). EGRIN 2.0 predicts, for example, that MetJ (GREs #19, #87) acts in conjunction with PurR (GRE #4) to differentially regulate genes specific to the pyrimidine biosynthetic branch (*pyrimidine* corem), while (yet to be identified) TFs that bind GREs #2 and #206 function with PurR (GRE #4) to regulate genes in the purine branch (*purine* corem) (Fig 4A). The organization of these GREs within and across promoters, and the environments in which they act to mediate regulation by specific TFs, generates complex co-expression patterns among different combinations of genes in the three corems of this highly canalized pathway (filled violin plots, Fig 4C). These conditional co-expression patterns predict that in certain environments, the two branches are differentially regulated, while in others they are co-regulated as one unit. Consistent with this observation, fitness consequences of deleting genes in these corems vary across conditions (Fig 4D, Supplementary Figs S28 and 29). For instance, knockouts of genes in all three corems have

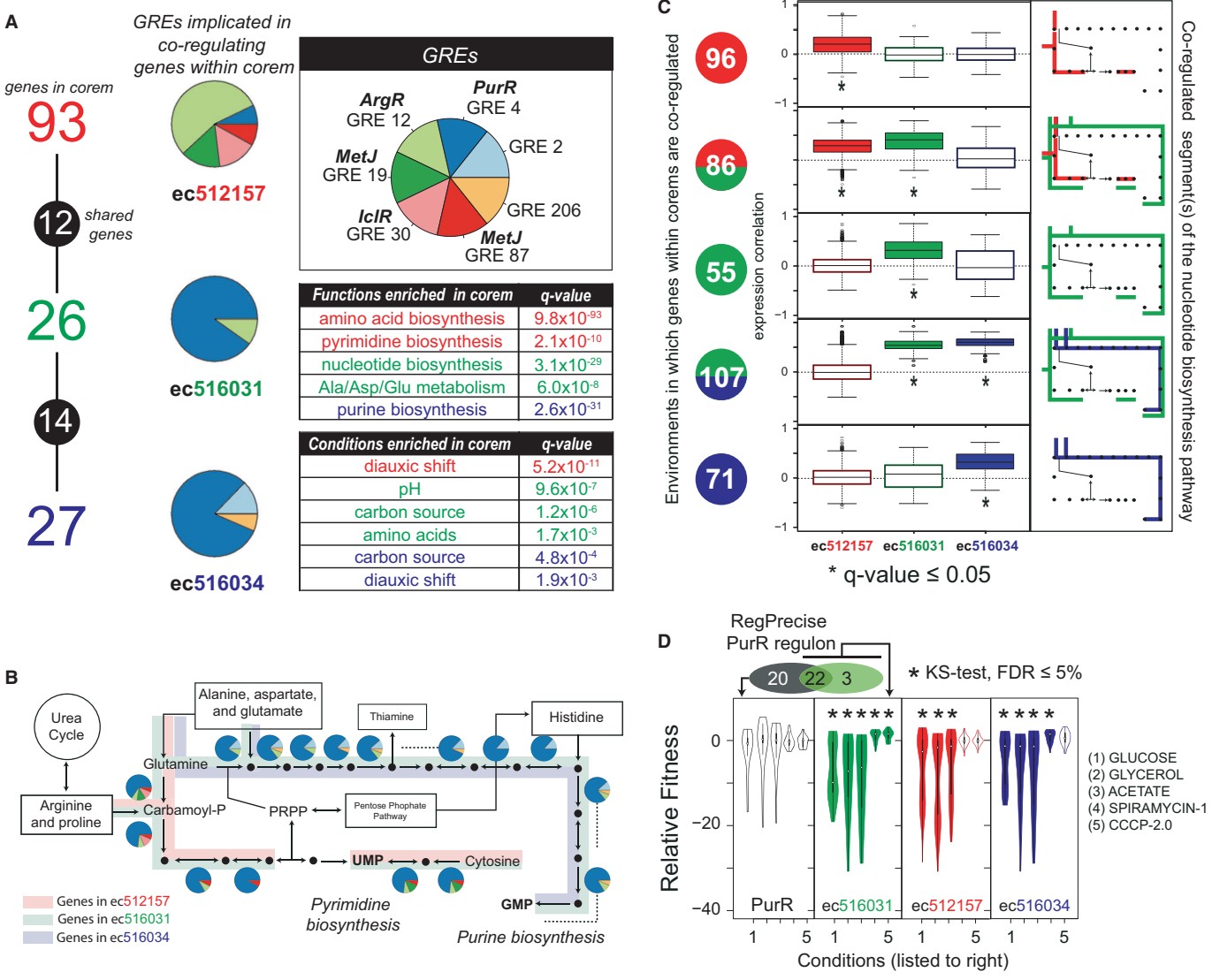

**Figure 4. Varying combinations of GREs act conditionally to subdivide and coordinate branches of the nucleotide biosynthesis pathway in an environment-dependent manner (*Escherichia coli*).**

Three corems model differential co-regulation of purine and pyrimidine biosynthetic genes in *E. coli*. Predictions are supported by expression data as well as fitness data.

A Genes of nucleotide biosynthesis are distributed in overlapping combinations across three *E. coli* corems: purine (ec*516034*—"*purine* corem"), pyrimidine (ec*512157*—"*pyrimidine* corem"), or both pathways (ec*516031*—"*nucleotide* corem"). (Left) Gene membership and overlap for the three corems as in Fig 3C. Pie charts indicate average GRE composition across all gene promoters in each corem (see Supplementary Fig S6 for detail). (Top-right inset) GRE key for pie charts. Matches to TFs in RegulonDB noted above the GRE name. (Bottom-right) Tables list enriched gene functions (Dennis *et al*, 2003) and environmental conditions for each of the corems (see Supplementary Information).

B A portion of the nucleotide biosynthetic pathways, near the branch point dividing purine (top) and pyrimidine (bottom) biosynthesis. Pie charts represent GRE composition in each gene promoter (key in A). Operons denoted by dashed lines, with only the leader gene's promoter architecture shown.

C Condition-specific co-expression of genes across the three corems. (Right) The active segments of nucleotide biosynthesis (as in B) are color-matched to corems. (Center) Box plots show distributions of expression correlations between genes within each corem in relevant environmental conditions, when they are predicted to be co-regulated. Color fill and asterisks indicate corems with significantly low relative standard deviation (RSD; $|\sigma/\mu|$; FDR ≤ 0.05). (Left) Colored circles indicate when genes within which corem(s) are predicted to be co-regulated (color) under how many conditions (number).

D Distributions of relative fitness values for gene deletions in the three corems, as well as 20 of the 42 PurR regulon genes not modeled by ec*516031* (black) across 5 representative conditions (condition identifiers listed to right, additional conditions in Supplementary Fig S29). Asterisks denote conditions in which the distribution of fitness values is statistically significant (relative to the distribution of fitness values for all genes in that condition).

similar consequences on fitness in the presence of glucose. By contrast, in the presence of the toxic ionophore carbonyl cyanide m-chlorophenyl hydrazone (CCCP), only knockouts of genes in the *nucleotide* corem significantly alter fitness.

This example highlights two important features of EGRIN 2.0 and corems. First, EGRIN 2.0 can distinguish co-regulation by independent, similarly acting TFs, even though their targets are co-expressed. Further, corems group together genes that are functionally related

even though their co-regulation is mediated by different mechanisms, demonstrating how conditional TF influences in a GRN coordinate transcription of genes from *different* regulons whose deletions have highly correlated fitness consequences (Table 1). Genes of the *pyrimidine* corem, for example, are co-regulated by as many as five TFs. Even though promoters of each of the genes in this corem contain distinct compositions of GREs (Supplementary Fig S27, Supplementary Datasets S8 and S9), their expression is highly coordinated across a broad range of conditions. Interestingly and counter to our expectation, transcript level changes of the similarly acting TFs are not highly correlated. Instead, we discovered correlated changes in the concentrations of effector molecules, which allosterically regulate the activities of these TFs, suggesting that coordinate regulation of genes in the *pyrimidine* corem is a direct consequence of metabolic dynamics (Supplementary Fig S30; Ishii *et al*, 2007).

Second, EGRIN 2.0 predicts that not all locations that match to the same GRE are functionally equivalent in all environments. Accordingly, using corems, we can discern and explain why genes regulated by the same TF exhibit different expression patterns in certain environments. For example, out of the 42 PurR-regulated genes (assigned by RegPrecise), expression changes of the 14 that are grouped into the *purine* corem are better correlated with each other and genes of this corem than they are to the portion of the PurR regulon that was left out (*t*-test, $P < 2.2 \times 10^{-16}$, Fig 5A). Consistent with this observation, PurR is predicted to play a variable role in the regulation of genes across the three corems (from being highly important for the *nucleotide* corem, to being marginally important for the *pyrimidine* corem, Fig 4A). We hypothesized that the degree to which PurR is implicated in regulating genes within each corem is a good predictor of target-specific expression consequences of knocking out this TF. To test this hypothesis, we analyzed global transcriptional changes in both wild-type (WT) and $\Delta purR$ deletion strains of *E. coli* grown in the presence of adenine (Cho *et al*, 2011). These data were obtained from experiments that were not included in the construction of the EGRIN 2.0 model. Specifically, we calculated the relative standard deviation (a measure of co-regulation) for every PurR-associated corem in each of the two strains. As expected, genes in all three corems described above were co-regulated in the WT strain (FDR < 0.05, Fig 5B). Strikingly consistent with EGRIN 2.0 predictions, the degree of dysregulation of genes within each of the three corems in the $\Delta purR$ strain was proportional to the predicted magnitude of PurR influence. Maximal dysregulation of genes in the *nucleotide* corem and the *purine* corem, for example, was consistent with the predicted role of PurR as the primary regulator of genes in these corems (Fig 5C). Notably, the degree of disruption observed in these two corems surpasses that of the entire PurR regulon, suggesting that in the presence of adenine, PurR regulates only a subset of its target genes. These results illustrate how the concept of a corem captures the context in which TF binding to a GRE is functional, not just that the potential for TF–GRE interaction exists, which is how a regulon is defined.

## Discussion

EGRIN 2.0 explains how microbes tailor transcriptional responses to varied environments by linking the genome-wide distribution of GREs to their organization and conditional activities within each promoter. The integrative model reveals the mechanisms by which microbes reuse genes in varying combinations to operationally link disparate processes and regulate flux through metabolic pathways. We have provided extensive validations for predictions made by EGRIN 2.0 for a bacterium and an archaeon (Table 2). In addition, we also performed new experiments to validate a model prediction that widespread transcriptional activity at non-canonical locations within genes and operons was partly responsible for complex modulation of the *E. coli* transcriptome during growth in rich media.

Corems represent a fundamental organizing principle of GRNs that captures fitness-relevant associations among genes, forging a link between the environment-dependent dynamics of transcriptional control and phenotype. The conditional associations among genes across corems reflect the underlying structure of coupled changes in environmental factors, such as correlated changes in effector molecules. Comparative analyses of EGRIN 2.0 models, therefore, could reveal the corems associated with unique and shared environmental structures that distinguish ecotypes of the same species.

Despite the vast amount remaining to be discovered about transcriptional regulation in even the most well-studied organisms, EGRIN 2.0 represents an important advance that may be useful for synthetic biology. Its usefulness for synthetic biology is twofold: (1) It opens the door for accurate and comprehensive inference of genome-scale models in any culturable organisms, and (2) it explicitly models the environmental dependence of regulatory mechanisms operating across the entire genome, including non-canonical locations. By teasing apart regulatory mechanisms that have indistinguishable outputs in some (but not all) environments, EGRIN 2.0 offers multiple strategies for introducing new genes into the GRN.

For instance, there are at least five distinct mechanisms responsible for co-regulating nearly 100 genes in the *pyrimidine* corem in *E. coli*. This corem coordinates genes from various segments of amino acid biosynthesis pathways, including arginine biosynthesis, as well as the pentose phosphate pathway to synchronize inputs into nucleotide biosynthesis. The conditional grouping of genes into the *pyrimidine* corem explains the previous observation that genes of arginine biosynthesis are repressed upon adenine addition (Cho *et al*, 2011). EGRIN 2.0 predicts that this coordination of nucleotide and arginine biosynthesis is accomplished by an equivalency of PurR and ArgR activities under these conditions (possibly due to correlated changes in effector molecules), rather than by direct regulation of arginine biosynthesis genes by PurR. Not surprisingly, subsets of genes within this corem belong to alternate regulatory programs (corems) under different environmental contexts. While the specific mechanisms that give rise to these nuanced, switch-like behaviors will need to be detailed by careful experimentation, one can imagine constructing a library of endogenously encoded coregulatory strategies based on *cis*-acting mechanisms that already exist within the GRN of an organism. Future work to translate the EGRIN 2.0 model into the language of synthetic biology will help enable system-level reengineering of an organism.

## Materials and Methods

Additional detail for each section provided in the Supplementary Information.

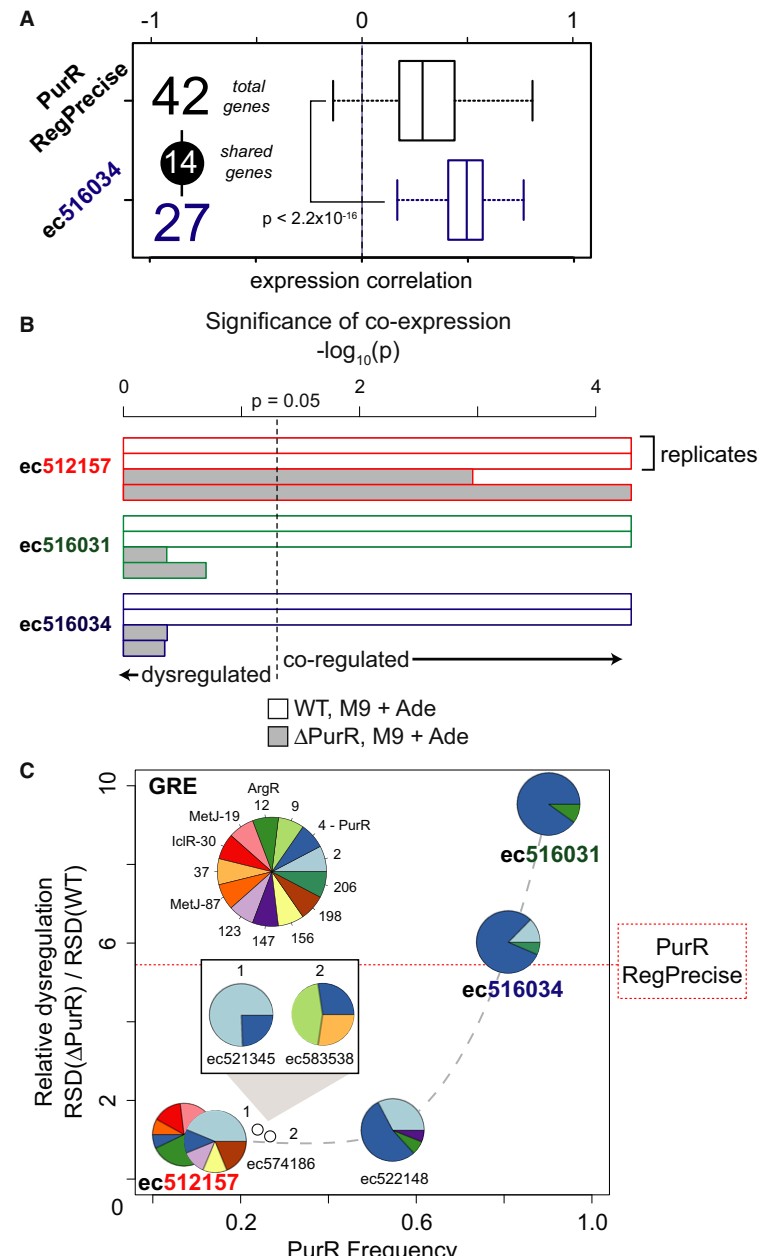

**Figure 5. EGRIN 2.0 predicts how conditional influences of a TF vary across all of its binding sites in the genome.**
Corems model a subset of genes from the PurR regulon that are tightly co-expressed and most affected by PurR knockout.

A   Distributions of pairwise expression correlations among all genes in the PurR regulon (RegPrecise) compared to a subset of the regulon within corem ec516034, across all environmental conditions. Also shown are the total number of genes in each group, and the number of shared genes. The two distributions are significantly different (Welch two-sample *t*-test, $P < 2.2^{-16}$).

B   RSD of transcript level changes (resampled -$\log_{10}(pval)$) for the three corems in Fig 4 in WT and $\Delta purR$ strains of *Escherichia coli* (both grown with adenine). The dashed line delineates significant co-expression ($P = 0.05$).

C   Relative RSD ($\Delta purR$/WT) for all seven GRE #4-associated corems plotted as a function of the frequency with which GRE #4 (PurR) is discovered within these corems. Composition of GREs discovered within each corem is shown as pie charts (as in Fig 4), with key in inset, top-right. Relative RSD of the RegPrecise PurR regulon is shown for reference (dotted horizontal line).

### Training data

#### Halobacterium salinarum NRC-1

Of 1,495 transcriptome profiles, *H. salinarum NRC-1* genome sequence (RSAT), STRING (Version 9).

#### Escherichia coli

Eight hundred and sixty-eight transcriptome profiles from (primary dataset; Lemmens *et al*, 2009) and 805 transcriptome profiles from DREAM5 (For comparison to RegulonDB only; Marbach *et al*, 2012), *E. coli* genome sequence (RSAT), STRING (Version 9).

**Table 2.  Summary of model predictions and experimental validations[a]**

| Prediction class | Specific Prediction | Validation | Location |
|---|---|---|---|
| Accuracy of *de novo* discovery of GREs | 337 GREs discovered and genome-wide locations predicted in *E. coli* | Predictions validated by genome-wide binding location data for 53 out of 88 characterized TFs | Fig 2A, Supplementary Dataset S2 |
| | Organization and composition of GREs within *H. salinarum kdp* promoter | *In vivo* transcription assays of truncated promoter constructs | Fig 2C |
| | Organization and composition of GREs within *E. coli carA* promoter | TF-binding locations within the *carA* promoter (RegulonDB) | Fig 2D |
| | ArgR and PurR binding sites in *E. coli pyrL* promoter | TF-binding locations mapped using ChIP-chip | Supplementary Fig S18 |
| Accuracy of TF–target interactions in the global GRN | Regulatory interactions between 132 TFs and 1,131 genes in *E. coli* | 555 interactions correct at 25% precision, RegulonDB | Fig 2A |
| Regulatory mechanisms at non-canonical promoter locations | 98 *H. salinarum* operons with condition-specific transcript isoforms | 40 confirmed by tiling array | Supplementary Figs S19, S20 and S21, Supplementary Dataset S6 |
| | 189 *E. coli* operons with condition-specific isoforms | 58 confirmed by tiling array | Supplementary Figs S11, S12 and S13, Supplementary Dataset S6 |
| | Conditional isoforms of *H. salinarum dpp* operon | Tiling array and binding locations for TFBs | Fig 3, Supplementary Figs S22, S23, S24 and S25 |
| Conditional regulation of branch points within metabolic pathways | Segmentation of nucleotide biosynthesis pathway by multiple TFs | Condition-specific co-expression; TF effector molecule correlation; Condition-specific fitness consequences of gene deletions | Fig 4, Supplementary Figs S26–S30 |
| Physiological consequences of deleting genes within corems | Corems establish a better relationship between co-regulation and fitness | Deleting genes within corems result in similar fitness consequences in chemical genomics screen | Fig 2B and Supplementary Fig S14 |
| | Similar action by different TFs results in co-regulation of genes across regulons | Highly correlated fitness effects of deleting genes within the same corem, albeit from different regulons | Table 1, Supplementary Fig S30, Supplementary Datasets S4 and S5 |
| Regulation by a TF varies conditionally across different targets in the genome | Degree of PurR influence on regulation of its target genes across different corems | Increased RSD in Δ*purR*/WT strains proportional to PurR influence | Fig 5C |

[a]All validations were performed with data from experiments that were not used in model construction.

Full description of each dataset including normalization and a breakdown of the composition of each dataset is provided in the Supplementary Information.

## Validation data

Eight independent datasets were used to validate model predictions. 4 out of 8 were generated in our laboratory. Validation data were not used for model training.

### *Halobacterium salinarum NRC-1*

High-resolution (12 nt) tiling array transcriptome measurements were collected over 12 points along the *H. salinarum* growth curve in rich media. These were published in a separate study (Koide *et al*, 2009). ChIP-chip binding profiles for eight general TFs and three specific TFs were collected from Facciotti *et al* (2007). NRC-1 *kdp* truncation data were obtained from Kixmuller *et al* (2011).

### *Escherichia coli*

Transcriptome profiles for *E. coli* using high-resolution (23 nt) tiling array were measured at nine different time points during growth in rich media (GSE55879).

Fitness measurements across 324 conditions were generated by Nichols *et al* (2011). PurR/ΔPurR expression data and ChIP-chip transcription factor binding measurements were collected from Cho *et al* (2011). Effector molecule measurements were supplied by Ishii *et al* (2007). All comparisons with RegulonDB were performed against version 7.2 of the database (Gama-Castro *et al*, 2011).

See Table 2 for complete list of validated predictions and references. Complete description of each validation dataset is provided in the Supplementary Information.

### EGRIN 2.0 construction

EGRIN 2.0 was constructed as an ensemble of many individual EGRIN models (~500 for *H. salinarum* and ~100 for *E. coli*). Each EGRIN model was constructed using two algorithms: *cMonkey* (Reiss *et al*, 2006), to learn condition-dependent modularity of the regulatory network, and *Inferelator* (Bonneau *et al*, 2006), to infer regulatory factors (transcription and/or environmental factors) influencing the expression of the modules. A full description of the cMonkey and Inferelator algorithms is provided in the Supplementary Information.

Following a basic model averaging approach (Breiman, 1996), we integrated the EGRIN models and mined the ensemble to discover frequently reoccurring features and associations (Fig 1D). Brief description of each step is provided below. Full description, including benchmarking, is provided in the Supplementary Information.

We refer to the modules detected by our procedure as co-regulated modules, or corems, the frequently re-occurring *de novo cis*-regulatory motifs as GREs, and the overall framework and model as *EGRIN 2.0* (see Materials and Methods, Supplementary Information, and Supplementary Fig S1 for a detailed workflow).

Ensemble statistics are provided in Supplementary Table S3. Full description of the algorithms and each post-processing step is documented in Supplementary Information. Below we summarize key steps.

### GRE discovery

Conserved *cis*-acting GREs discovered in biclusters (MEME; represented as position-specific scoring matrices, or PSSMs) were aligned and compared using *Tomtom* to compute pairwise similarities (Euclidean distance, Gupta *et al*, 2007). The resulting network of highly similar PSSM pairs was clustered using *mcl* (FDR ≤ 0.01 and overlap of 6 nt, Van Dongen, 2008). Cluster containing at least 10 PSSMs were considered gene regulatory elements or GREs (Supplementary Datasets S1 and S2). Combined PSSMs for each GRE (e.g., Fig 2E, Supplementary Fig S2) were computed as the unweighted mean of aligned PSSMs within each cluster.

GRE locations throughout the genome were computed using *MAST* (Bailey & Gribskov, 1998), subject to a *q*-value threshold of 0.01 for alignment of each PSSM within a GRE at each genomic location. Motif counts (e.g., Figs 2C–E and 3A) were computed by summing significant matches at each genomic locus.

### GRE-TF matching

GREs were matched to TFs by comparing their genomic locations to binding sites for all experimentally characterized TFs in RegulonDB (*E. coli*; *BindingSiteSet* table, filtered for experimental evidence and TFs with three unique binding sites; a total of 88). A GRE was considered a significant match to a TF if a significant fraction of PSSMs in the GRE had genomic locations that significantly overlap with the experimentally mapped binding sites for the TF (FDR ≤ 0.05 and *P*-value ≤ 0.01, respectively). In the case that a GRE matched multiple TFs, only the most significant TF match was retained. We note that in some cases, multiple GREs can also match a single TF (additional details provided in the Supplementary Information).

### Corem detection

We transformed the EGRIN 2.0 ensemble into a gene–gene association network by ranking the frequency with which each pair of genes co-occurred among all biclusters. We removed associations that were indistinguishable from noise using network backbone extraction (Serrano *et al*, 2009). Finally, we computed conditionally co-regulated modules, or *corems,* using link-based community detection algorithm (Ahn *et al*, 2010). Since corems were defined as links between genes, a given gene can be a member of multiple

communities. Corem statistics are provided in Supplementary Table S3. Corem–gene memberships are provided on the Web site.

### Deciphering environmental context and GREs responsible for co-regulation of corems

We considered a corem to be co-regulated in experimental conditions where the relative standard deviation (RSD = $|\sigma/\mu|$) among genes in the corem was significantly low (permutation $P \leq 0.05$). We implicated GREs for conditional co-regulation of a corem if they were: (1) located within a 1,000 nt window ($-875$ nt to $+125$ nt) around the start codon of any gene in the corem; and (2) frequently discovered in biclusters containing genes from the corem (*i.e.*, top 10% of biclusters, ranked by number of corem genes in the bicluster).

### Annotation of environmental context

Extensive metadata collected about each experiment for *H. salinarum* was collated into an 'environmental ontology' that formalizes the hierarchical relationships between experimental conditions. The environmental ontology was used to annotate conditions in *H. salinarum* throughout the manuscript. The ontology is available on the supporting Web site.

### Comparison with *DREAM5*

To compare EGRIN 2.0 performance with DREAM5, we computed an EGRIN 2.0 ensemble on the dataset described in Marbach *et al* (2012). We subdivided the *E. coli* EGRIN 2.0 model into two predicted GRNs: (1) a "direct" GRN (based upon *Inferelator* predictions) and (2) a "GRE-based" GRN that was computed by matching *E. coli* TFs to GREs (described above). We used the published DREAM5 ensemble predictions (Marbach *et al*, 2012). All GRNs were compared using the RegulonDB gold-standard curated by Marbach *et al*. The gold-standard includes 2,066 interactions classified with a "strong evidence" code in RegulonDB. Precision-recall curves and AUPR statistics were calculated as described in Marbach *et al* (2012).

### False detection rates

We used the Benjamini–Hochberg procedure for significance assessments of findings that required correction for multiple comparisons. Individual and collective corrected *P*-values are reported as *q*-values and false discovery rates (FDR), respectively.

**Supplementary information for** this article is available online: http://msb.embopress.org

### Acknowledgements

This work conducted by ENIGMA was supported by the Office of Science, Office of Biological and Environmental Research, of the U.S. Department of Energy under Contract No. DE-AC02-05CH11231. Additional funding was provided by grants from the U.S. Department of Energy (DE-FG02-04ER64685 to NB, DE-FG02-07ER64327 to NB, DE-FG02-08ER64685 to NB); the U.S. National Science Foundation (EAGER—MSB-1237267 and Interplay—NSF-1330912 to NB and ABI—NSF-1262637 to NB and DJR); the U.S. National

Institutes of Health, Center for Systems Biology (2P50GM076547 to NB); and by the University of Luxembourg-ISB partnership. ANB supported by the Department of Energy Office of Science Graduate Fellowship Program (DOE SCGF), made possible in part by the American Recovery and Reinvestment Act of 2009, and administered by ORISE-ORAU under contract no. DE-AC05-06OR23100. DMS supported by São Paulo Research Foundation (FAPESP) grants 2012/05392-1 and 2011/08104-4. We thank Justin Ashworth, Adrián López García de Lomana, Ben Heavner, James Eddy, and Serdar Turkarslan for helpful comments.

## Author contributions

DJR, ANB, and NSB conceived the study. DJR ran cMonkey and Inferelator and constructed the ensemble. ANB and AA performed corem analyses. DJR and ANB analyzed the data and performed external validations. SC provided assistance with RegulonDB comparisons; CLP provided assistance with GRE clustering. ANB designed and supervised experimental validations. AK and MP conducted experiments. ANB, DMS, and WW developed the Web resource. ANB, DJR, and NSB wrote the paper. NSB and DJR supervised the study. All authors have read and approved the manuscript.

## Conflict of interest

The authors declare that they have no conflict of interest.

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
