## [Review Process File · Molecular Systems Biology]

A system-level model for the microbial regulatory genome

Aaron N Brooks, David J Reiss, Antoine Allard, Wei-ju Wu, Diego M Salvanha, Christopher L Plaisier, Sriram Chandrasekaran, Min Pan, Amardeep Kaur and Nitin S Baliga

Corresponding author: Nitin Baliga, Institute for Systems Biology

Review timeline:

Submission date:	28 January 2014
Editorial Decision:	20 March 2014
Revision received:	27 May 2014
Editorial Decision:	06 June 2014
Accepted:	11 June 2014

Editor: Maria Polychronidou

Transaction Report:

1st Revision - authors' response

20 March 2014

Thank you again for submitting your work to Molecular Systems Biology. I would like to apologize for the delay in getting back to you. We have now heard back from the three referees who agreed to evaluate your manuscript. As you will see from the reports below, the referees acknowledge that you address a potentially interesting topic. However, they raise a series of concerns, which should be carefully addressed in a revision of the manuscript.

Without repeating all the points listed below, among the more fundamental issues are the following:

- A more thorough comparison of the EGRIN2.0 performance to that of existing approaches needs to be included.
- Independent experimental validation would be required for better supporting the newly predicted associations.
- Potential issues arising from normalization when integrating different datasets need to be carefully addressed.

Moreover, several of the referees' comments refer to the need to clarify and better document several points throughout the manuscript. As referee #1 mentions, all methods, algorithms, datasets etc. need to be clearly documented while avoiding repetitions in the main text and supplemental information.

On a more editorial level, while we appreciate that you introduce the new term 'corem', we would strongly recommend avoiding the use of jargon throughout the manuscript and we would suggest using the term 'co-regulated modules' instead. Additionally, we would like to ask you to provide the Supplementary Figures in individual files, each containing the Supplementary Figure and the corresponding Supplementary Figure legend.

If you feel you can satisfactorily deal with these points and those listed by the referees, you may wish to submit a revised version of your manuscript. Please attach a covering letter giving details of the way in which you have handled each of the points raised by the referees. A revised manuscript

will be once again subject to review and you probably understand that we can give you no guarantee at this stage that the eventual outcome will be favourable.

REFEREE REPORTS

Reviewer #1:

Overview

In this work from the Baliga lab, Brooks et al. describe the extension of their EGRIN model from 2007 and the results that they have obtained by applying the EGRIN 2.0 methodology in two model organisms, *E. coli* and *H. salinarium*. The core of the new EGRIN 2.0 framework is the discovery of "corems", which are (overlapping) sets of co-regulated genes whose activity/function is condition-specific. The authors demonstrate the utility of their technique by describing specific corems and discussing the validation tests that they performed for a number of prediction classes.

As a general comment, the manuscript is well written, both main and supplementary text have correct annotation and the right level of detail to ensure proper dissemination of the results. The figures are of high quality and convey the message. I found them to be too convoluted (e.g. violin plots) in certain cases and I would encourage the authors to clarify, even more so, what the important result in each case is.

Specific comments:

1. Methods: the manuscript and supplementary methods are quite redundant in the information that they have. In addition, the suppl. methods do not have all the information necessary to understand the knowledgebase used and algorithms constructed. I would suggest the authors build more upon their efforts and create an all-inclusive suppl. methods section (currently some info is missing, for example # of *E. coli* arrays in dataset, although it exists in main manuscript) and succinctly describe these efforts in the main manuscript (not duplicate).

2. Suppl. p5:21: Be more descriptive of the *E. coli* dataset that was used, what kind of data that was there (mention it as "the *E. coli* data set" gives no information), number of arrays, what kind of normalization was used, what is the distribution of the various experimental conditions (KOs, carbon sources, etc.). This information is paramount for (a) confirming that bias is removed from the training set and (b) understanding what experiments are covered and thus a predictive model can learn upon to generalize to other conditions.

3. The issue of normalization warrants a closer look. The authors "collate" expression datasets from different platforms, labs, databases, etc. This will definitely introduce bias on the final dataset and the algorithms will predict associations between variables (genes) that are otherwise conditionally independent. It seems that the authors did not consider batch effects and normalization issues in this work, which is a major omission. I would like to see this issue addressed, with clear quality control plots (before/after normalization) and the methodology retrained on the new dataset. Depending on the bias, the "corem" membership might also change substantially.

4. The network inference validation part of the paper has room for improvement. The authors try to compare to CLR and DREAM5 but why only CLR and not top performers, such as GENIE3, TIGRESS, ANOVA, etc. is not clear). In addition, for DREAM5 the authors took the predictions that were published in the Maybach 2012 paper, that were found by using a different expression dataset and using a similar but substantially different golden standard (mixed weak and strong interactions from a previous RegulonDB, v6.8). The authors will have to generate the consensus network by running the DREAM5 ensemble (available online as the GenePattern suite, also individual codes). If this is not doable because of the size (unlikely), they can also run independently the top (3-5) performers from each type of inference method from the Maybach paper to create the ensemble. That would be a valid comparison with the state-of-the-art. In addition, I would like to see how EGRIN 2.0 performs at golden standards of increasing strength in terms of experimental evidence (e.g. confirmed interactions vs. strong vs. weak; see RegulonDB).

5. Other questions that I have regarding the validation method: what are the negatives in the golden

standard? Should I guess that they are ALL other combinations between TFs and all other genes? That would produce a very biased testing set with hundreds of thousands negatives, many of which are false negatives. Is that reasonable? Why is AUPR calculated at 0.25 precision (I understand this is what Maybach et al. did, but this is not a justification by itself)? I would also like to see the ROC/PR curves (as suppl. figures) and the authors to confirm that they have constructed a convex hull when calculating the AUPR and not just sampled (otherwise results can differ substantially).

6. The authors have to perform cross-validation (which is noticeably absent in all testing performed) that would be essential to assess the generalization error of their techniques (together with the points in (5)).

7. I think that the specific examples they provide (nucleotide biosynthesis) are logical and interesting. Table 2 is a great summary of the validation/datasets/results. Something that is not crystal clear from the paper (wording mainly) is whether the authors just split the existing, previously published data to testing/training or they actually performed experimental validation for a subset of these cases. If the former is the case, to be considered for publication to MSB, I would like to see independent experimental validation for the newly predicted associations for a sufficient subset of the predictions. For Maybach et al. that number was a couple hundred interactions, here it can be the top 15-20 predictions with standard methods (ChIP-Seq, Y1H).

Conclusion:

This is an interesting, well-written paper that explores an important problem regarding context-dependent co-regulation with some major issues that have to be addressed before it can be considered for publication.

Reviewer #2:

This paper is an interesting development of previous bi-clustering and inferitor methodologies. By using a combination of bi-clustering, correlation analysis, sequence motifs and others they managed to identify groups of correlated genes under defined conditions, as well as identify putative regulatory sequences. I find the paper interesting but I have some questions I would like the authors to answer;

-How were the TFs defined? Based on database analysis, or where they inferred from the analysis? If they used ones from database, how they can be sure they did not miss any?, if from database which ones?. What will happen if they randomly select ten enzymes and define them as TFs, will they find GREs and assign them regulatory regions?. In other words how the method is dependent on having the TFs previously identified?

How many of the TFs binding logos they could recapitulate from the analysis if this info is not provided? What happens with TFs that don't change expression but are regulated by metabolites or posttranslational regulation? Out of the 40% TFs they could not find the binding sites, how many are non-dynamic?,

How do the authors explain cores? ie groups of genes regulated the same way but by diff TFs? Are the diff TFs regulated the same way? could more general effects take place, like supercoiling, chromatin organization?

They also mentioned that many TF binding sites are not operational under certain conditions, how do they explain these cases in bacteria?

The author say that their analysis will be extremely useful for synthetic biology, but I am doubtful. They find many nice correlations, they identify regulatory regions, but they cannot explain many of the observations (ie genes co-regulated although by diff TFs, binding sites not used etc...).

In summary I think it is a nice story, but I would like a discussion of the limitations of the methodology.

Reviewer #3:

This paper combines information from genomic sequence and expression data to generate a map of co-regulated gene expression modules. In the paper the method is applied/validated to *E. coli* and *H. Salinarium*; applicability to other (sequenced) microbes makes this paper interesting for a broad audience.

The authors show convincingly that the combination of several pre-existing methods for gene network reconstruction yields improved predictions of network organization, transcript isoforms, and promoter structure.

Validation of model predictions with experimental data and comparison with existing algorithms is performed adequately throughout all subsections.

The paper is clearly structured and well written, but in parts relies heavily on previous publications on EGRIN/cMonkey/inferelator making some sections difficult to understand for readers not too familiar with those papers.

Minor point: from the figure legends it remains unclear what the difference between data displayed in figures 3C and 4A is.

1st Revision - authors' response

27 May 2014

Reviewer #1:

Overview

In this work from the Baliga lab, Brooks et al. describe the extension of their EGRIN model from 2007 and the results that they have obtained by applying the EGRIN 2.0 methodology in two model organisms, E. coli and H. salinarium. The core of the new EGRIN 2.0 framework is the discovery of "corems", which are (overlapping) sets of co-regulated genes whose activity/function is condition-specific. The authors demonstrate the utility of their technique by describing specific corems and discussing the validation tests that they performed for a number of prediction classes.

As a general comment, the manuscript is well written, both main and Supplementary text have correct annotation and the right level of detail to ensure proper dissemination of the results. The figures are of high quality and convey the message. I found them to be too convoluted (e.g. violin plots) in certain cases and I would encourage the authors to clarify, even more so, what the important result in each case is.

RESPONSE:

We thank the referee for succinct description of our work and its impact, as well as the kind words regarding presentation of the data. It took great effort to make these complex results accessible to broad audience and we are glad to see that the central message can be grasped readily.

We appreciate the reviewer's concern about the amount of information conveyed in the figures. We acknowledge that violin plots contain more visual information than alternative representations (e.g., box plots) and should, therefore, be used judiciously. We have replaced the violin plots in Figure 4C with boxplots and agree that this helps clarify the central message of the figure. We contend, however, that violin plots are necessary for interpreting the data presented in Figure 4D. In this case, the distributions are not normal - meaning a box plot would misrepresent the true distributions (thus the reason that we use nonparametric statistics to evaluate its significance). A violin plot best conveys this information, where the reader has visual access to the complex relationship among fitness values.

To address the reviewer's concern about summarizing the central message of

each figure, we have clarified the central result conveyed by each figure. We have done so by including one or two sentences summarizing each figure in the caption, e.g. Page 25: Line 25, Page 26: Lines 17-19, Page 27: Lines 7-8, Page 27: Lines 39-40.

Specific comments:

1. *Methods: the manuscript and Supplementary methods are quite redundant in the information that they have. In addition, the suppl. methods do not have all the information necessary to understand the knowledgebase used and algorithms constructed. I would suggest the authors build more upon their efforts and create an all-inclusive suppl. methods section (currently some info is missing, for example # of E. coli arrays in dataset, although it exists in main manuscript) and succinctly describe these efforts in the main manuscript (not duplicate).*

RESPONSE:

We acknowledge that some information is duplicated between the main text and the **Expanded View***, while some relevant details were omitted from the **Expanded View** (e.g. by including references to prior publications). We have moved many details to the **Expanded View**, cleaned up and simplified the methods in the main text to include only the primary information, and greatly expanded the description of the methods and data sets in the **Expanded View**. Because the updated **Expanded View** is now significantly longer, we have added a table of contents and re-organized the text to improve its use as a reference. We also connect the reader to an online resource, which allows them to explore details that would be difficult to include in a static, written document. These modifications to the **Expanded View** make the algorithm, methods, and validations far more transparent. We thank the reviewer for encouraging us to make these improvements.

- ‘Expanded View’ is the MSB name for the Supplementary Materials

2. *Suppl. p5:21: Be more descriptive of the E. coli dataset that was used, what kind of data that was there (mention it as "the E. coli data set" gives no information), number of arrays, what kind of normalization was used, what is the distribution of the various experimental conditions (KOs, carbon sources, etc.). This information is paramount for (a) confirming that bias is removed from the training set and (b) understanding what experiments are covered and thus a predictive model can learn upon to generalize to other conditions.*

RESPONSE:

We agree that thorough description of the data sets is critical to understand, assess, and build from our work. Since another group generated the dataset, we chose to refer readers to their paper, which has information about the data set. However, based upon the referee’s suggestions (as highlighted above as well), we have now included an expanded description of the data set directly from the Lemmens *et al.* paper. The expanded description includes a breakdown of the experimental conditions, e.g. (Section 2.1.2, **Expanded View**) “The experiments cover a range of conditions, including varying carbon sources (136 arrays), pH (46 arrays), oxygen (284 arrays), metals (27 arrays) and temperature (23 arrays).” We thank the referee for noting this omission, which could hinder independent evaluation or re-implementation of our methods.

3. *The issue of normalization warrants a closer look. The authors "collate" expression datasets from different platforms, labs, databases, etc. This will definitely introduce bias on the final dataset and the algorithms will predict associations between variables (genes) that are otherwise conditionally independent. It seems that the authors did not consider batch effects and normalization issues in this work, which is a major omission. I would like to see this issue addressed, with clear quality control plots (before/after normalization) and the methodology retrained on the new dataset. Depending on the bias, the "corem" membership might also change substantially.*

RESPONSE:

The reviewer makes an important observation about normalization. We have (in part) answered this question for *E. coli* in the previous response (and we provide additional detail below). We reiterate that the *E. coli* data comes from a previously published data set. In short, we can demonstrate that we have followed standard normalization procedures for our data and prove that our results are consistent across two data sets normalized in different ways and by different groups.

For the *H. salinarum* model, we collected all of the data ourselves using two tightly-controlled platforms (two-color cDNA arrays, either spotted or tiling). To maintain internal consistency between the arrays collected in our lab, we measured an internal reference RNA on every slide (collected at mid-log growth; including dye-flip). Each batch of reference RNA was compared to the previous batch to maintain consistency. While these procedures have been described previously (Bonneau et al. 2007), we have now added a detailed description of the methods in the updated **Expanded View** to remind readers of our normalization procedures (Section 2.1.1). Prior to running cMonkey, the entire expression compendium is standardized on a per gene basis (mean removed and variance scaled), effectively creating z-scores for each gene.

For *E. coli*, we used a data set previously prepared by Lemmens, *et al.* (2009). Since the normalization for this data was fully described in their paper, we had simply referenced it. However, given the justified concern about the data normalization, we have included a summary of their normalization methodology in the revised **Expanded View** (Section 2.1.2.1).

More importantly, we have now validated our predictions in an independent expression compendium (**Figure E17**; the DREAM5 data set, Section 6.3 **Expanded View**). This data set has a different composition of arrays and alternate normalization procedure (summarized in Section 2.1.2.2 **Expanded View**). Since we find support for nearly 99% of the corems in both data sets (discussed in detail below, Reponse 6), we conclude that data normalization is not a significant concern for the results that we have presented.

4. The network inference validation part of the paper has room for improvement. The authors try to compare to CLR and DREAM5 but why only CLR and not top performers, such as GENIE3, TIGRESS, ANOVA, etc. is not clear). In addition, for DREAM5 the authors took the predictions that were published in the Maybach 2012 paper, that were found by using a different expression dataset and using a similar but substantially different golden standard (mixed weak and strong interactions from a previous RegulonDB, v6.8). The authors will have to generate the consensus network by running the DREAM5 ensemble (available online as the GenePattern suite, also individual codes). If this is not doable because of the size (unlikely), they can also run independently the top (3-5) performers from each type of inference method from the Maybach paper to create the ensemble. That would be a valid comparison with the state-of-the-art. In addition, I would like to see how EGRIN 2.0 performs at golden standards of increasing strength in terms of experimental evidence (e.g. confirmed interactions vs. strong vs. weak; see RegulonDB).

RESPONSE:

The reviewer makes several important points regarding comparison of our methods to DREAM5 using the RegulonDB gold standard. We have made several changes to the manuscript to make this more accurate, as suggest by the reviewer. Specifically, we have rerun our algorithm on the DREAM5 data set and used the published DREAM5 gold standard directly for our comparisons. The comparisons we report are from this updated analysis (e.g., Figure 2A, Table E4). We are grateful for the reviewer's comments, which have strengthened our comparison.

Since the Marbach *et al.* (2012) manuscript already compares performance of the DREAM5 community ensemble to the top individual performers; however, we

feel that a direct comparison of our methods to the DREAM5 ensemble is most appropriate (i.e., rather than comparing our method to the algorithms in DREAM5 individually). In addition, we were unsure about the utility of comparing our method to the gold standard across different evidence codes in RegulonDB - especially since this was not done in Marbach *et al.* for *E. coli*. This means that we would have no comparison for our results and, more importantly, these data are not provided in the DREAM5 gold standard (meaning we would once again have to use a different validation data set). As we elaborate below, our intention with the comparison is not to say that our method is “better” than other published approaches. Rather, we wish to highlight the power of training on a different type of information (in this case *de novo* detected *cis*-regulatory motifs). Our comparison demonstrates that training on additional information improves the precision-recall performance of our method, even with respect to components of our own methodology (i.e. Inferelator, which we note is one of the algorithms in the DREAM5 ensemble).

The DREAM5 ensemble is an important benchmark against which all future methods can be assessed. Rather than run DREAM5 methods on our data set as suggested, however, we decided to rerun our algorithm on the DREAM5 data set. We did this for three reasons: (1) we were not able to run their methods on our data set in a reasonable amount of time through the web interface (or download the methods to run locally), (2) we were concerned about selecting the correct parameters and thresholds for the various algorithms in DREAM5, and, most important, (3) we wanted to use this as an opportunity to assess the generalizability of our methods on a new data set (addressed in following responses).

To recap: we have recomputed our performance using the DREAM5 expression data set and gold standard directly. Note: these performance results are from an EGRIN 2.0 model trained on the DREAM5 data directly (a different expression compendium). The results and interpretation thereof are nearly identical to the originally reported comparison (e.g., 555 validated interactions at 25% precision compared to 577 reported originally). We have included the entire DREAM5 gold standard, full precision-recall curves, and each of the inferred networks in Table E5. We have also included precision-recall curves in Figure E8.

5. Other questions that I have regarding the validation method: what are the negatives in the golden standard? Should I guess that they are ALL other combinations between TFs and all other genes? That would produce a very biased testing set with hundreds of thousands negatives, many of which are false negatives. Is that reasonable? Why is AUPR calculated at 0.25 precision (I understand this is what Maybach et al. did, but this is not a justification by itself)? I would also like to see the ROC/PR curves (as suppl. figures) and the authors to confirm that they have constructed a convex hull when calculating the AUPR and not just sampled (otherwise results can differ substantially).

RESPONSE:

The reviewer provides criticisms regarding validation of the method using RegulonDB. We emphasize that we have used the same validation technique as performed by Marbach *et al.* for DREAM5.

The primary reason we use precision-recall is precisely to avoid the problem with ‘negatives’ mentioned by the reviewer. True negatives (TN) are especially problematic because we do not know if negatives in RegulonDB are really negative, or just haven’t been observed yet. Unfortunately, TNs are a critical component of ROC analysis through the False Positive Rate (FP/(FP + TN), where FP is the number of false positives). If we had used ROC, we could have artificially increased our score by, for example, making fewer predictions (especially where the TF-gene interaction data is sparse). We have avoided this by using precision-recall. Given data limitations (i.e., we cannot be sure that negatives are really negative), we believe our approach is the most conservative. As the reviewer notes, the choice of recall at 0.25 precision is arbitrary. We

provided it to make comparison of our method with DREAM5 easier. We find this to be a more tangible metric for a casual reader (compared to the AUPR; i.e., how many predictions are made when at least 1/4 of them are known). We do, however, understand the concern that we could have chosen a value that artificially inflates our performance. To address the reviewer's concern we now include recall at 10%, 25%, and 50% precision in **Figure 2B**. In addition, we have included the entire precision-recall curves in the **Expanded View** (Figure E8, Table E4).

6. The authors have to perform cross-validation (which is noticeably absent in all testing performed) that would be essential to assess the generalization error of their techniques (together with the points in (5)).

RESPONSE:

The reviewer's raises a concern regarding overfitting. The claim that crossvalidation was not performed, however, is not entirely accurate. In the following response we detail how our method inherently applies cross-validation, and, furthermore (and more important), we demonstrate that our observations are generalizable across data sets.

The primary motivation for using an ensemble approach was to avoid overfitting by deriving a consensus model. In other words, EGRIN 2.0 predictions are derived from reproducible predictions on individual models generated on subsets of the available data (each model being trained on roughly 1/4 of the available data, i.e., bagging). This is very similar to cross-validation. We have added additional details to the **Expanded View** to make this more clear (Section 4.3).

We have also included additional analysis to demonstrate that variance of the method decreases as additional EGRIN models are included in the ensemble (Figures E9, E15-16). We did this two ways: (1) We computed variance in the model's prediction performance (with respect to AUPR on RegulonDB gold standard, Figure E9) and (2) we calculated the similarity of the inferred networks as a function of the number of EGRIN models added to the ensemble (Figure E16). As expected, model variance decreases while predictive performance increases as a function of the number of runs included. At ~60 runs, for example, there is very little variance in the performance or underlying inferred networks (i.e. the ensemble has converged to a consensus model). Ensemble models are known for exactly this property (G. Seni, 2010, Ensemble Methods in Data Mining: Improving Accuracy Through Combining Predictions). This provides evidence that we have not overfit to particular measurements in the data set.

Another interpretation of the reviewer's generalization concern is that we have overfit the entire data set, i.e. our results may not hold in an independent data set. This concern is related to the reviewer's previous question regarding normalization of the training data set. To address this concern, we assessed whether corems predicted by the *E. coli* EGRIN 2.0 model (based upon the DISTILLER expression compendium) are supported in the independent DREAM5 expression data compendium. Further, we tested whether corems predicted to be highly condition-specific in the EGRIN 2.0 model are also observed to be condition-specific in the DREAM5 data set. We computed the condition-specific activity of the corems in both data sets. We found that nearly 99% of *E. coli* corems described in the manuscript are co-regulated in both data sets, i.e., the corem has significantly small coefficient of variation in at least one experiment (computed by resampling). More important, we confirmed that corems have similar condition-specific properties in both data sets, i.e. highly condition-specific corems in the DISTILLER data set are likely to be tightly co-expressed in few conditions in the DREAM5 data set while corems co-expressed in most conditions in DISTILLER are likewise co-expressed broadly in the DREAM5 data set (Figure E17, Spearman correlation of 0.49, pvalue < 1x10⁻⁶), after removing the intrinsic relationship between the number of genes in a corem and the number of conditions in which the corem is co-regulated by partial correlation).

This observation leads us to conclude that our model's predictions are generalizable across data sets. We have described this analysis in detail in the updated **Expanded View Section 6.3**.

7. I think that the specific examples they provide (nucleotide biosynthesis) are logical and interesting. Table 2 is a great summary of the validation/datasets/results. Something that is not crystal clear from the paper (wording mainly) is whether the authors just split the existing, previously published data to testing/training or they actually performed experimental validation for a subset of these cases. If the former is the case, to be considered for publication to MSB, I would like to see independent experimental validation for the newly predicted associations for a sufficient subset of the predictions. For Maybach et al. that number was a couple hundred interactions, here it can be the top 15-20 predictions with standard methods (ChIP-Seq, YIH).

RESPONSE:

We appreciate the referee's concern for experimental validation. Indeed, one of the reasons we constructed an *E. coli* EGRIN 2.0 model was to have access to many varied, independent experimental data sets that would provide validation even beyond TF-gene interactions. We agree that improving wording in the manuscript would help clarify which of the data sets were used as independent, experimental validations. We thank the Reviewer for bringing this to our attention. We have restructured the **Methods** section in the main text and the **Expanded View** to clearly distinguish between training and validation data.

We have used **8 independent experimental resources** to validate our model's predictions in *E. coli*. These validations are diverse - from TF-gene associations to detailed transcriptome structures to gene co-fitness relationships. Four of these validations were generated in our lab. To be clear - none of these data sets was used for model training. All of these experiments were either conducted in our lab or are re-analysis of raw data from other labs. We thoroughly documented and clearly separated the training and validation data in the revised **Expanded View**.

In particular, we would point the referee to the high-resolution *E. coli* transcriptome we generated to substantiate the model's prediction about transcriptional heterogeneity in *E. coli* (description beginning on Page 11 Line 30). To our astonishment, the model predicted that nearly 1/4 of all operons in *E. coli* have condition-specific isoforms. This would mean that the *E. coli* transcriptome is far more complex than has been appreciated. Using a high-resolution tiling array to measure gene expression across 9 time-points spanning the *E. coli* growth curve, we were able to *globally* validate the model's claim (pvalue = 1.07×10^{-5} , Section 5.3 **Expanded View**). We remind the reviewer that transcriptome structure was not in the training data; nevertheless, this was a precise, meaningful, and testable prediction made by the model that we confirmed through experimentation.

In addition, corems modeled 319 pairwise associations among genes from different regulons, whose deletions had similar fitness consequences across diverse environments (Pearson correlation ≥ 0.75). This is a remarkable finding that directly ties co-regulation to its downstream functional and phenotypic consequences (e.g., fitness). Discovery of these relationships is surprising given that fitness consequences were not a part of the training data. To our knowledge, EGRIN 2.0 is the first model to demonstrate such strong links between the complexities of co-regulation and fitness.

It is our view that the ability to validate higher-order organizational features of the microbial regulatory genome -- rather simply enumerate additional TF-gene associations -- is a key strength of our approach (although we were able to demonstrate TF-gene interactions that were not present in RegulonDB using ChIP-chip as well; see *pyrL* example, Figure E18, Table 2).

Conclusion:

This is an interesting, well-written paper that explores an important problem regarding context-dependent co-regulation with some major issues that have to be addressed before it can be considered for publication.

RESPONSE:

We thank the reviewer for insightful comments that have led to substantial improvements to this manuscript.

Reviewer #2:

This paper is an interesting development of previous bi-clustering and inferator methodologies. By using a combination of bi-clustering, correlation analysis, sequence motifs and others they managed to identify groups of correlated genes under defined conditions, as well as identify putative regulatory sequences. I find the paper interesting but I have some questions I would like the authors to answer;

-How were the TFs defined? Based on database analysis, or where they inferred from the analysis?, If they used ones from database, how they can be sure they did not miss any?, if from database which ones?. What will happen if they randomly select ten enzymes and define them as TFs, will they find GREs and assign them regulatory regions?. In other words how the method is dependent on having the TFs previously identified?

RESPONSE:

The referee notes an important point regarding annotation of transcription factors. We regret that we omitted this important detail from the **Expanded View**.

For the *E. coli* model we used a list of 296 *E. coli* putative TFs published by Marbach et al., 2012. We did this to enable direct comparison of our methods to the DREAM5 community network.

For *H. salinarum* we used a list of putative TFs collated by our lab. Our annotations are based on sequence (including predicted structural homology with known DNA binding domains).

We have included this important detail in Section 2.2.3 of the Expanded View, as well as including the complete list of TFs used for both organisms (available online).

As the reviewer notes, it is possible that our analysis has missed TFs (and included genes that are not actually TFs). The reviewer is correct that this will lead to errors. Unfortunately, these limitations are a source of error for any data-driven approach. This error is difficult to quantify because the number of regulatory proteins in any organism (even *E. coli*) is unknown.

We note, however, that we would not assign a GRE or regulatory region to proteins that are not *bona fide* TFs. Since our method relies on experimental evidence (e.g. ChIP-chip in RegulonDB) to map GREs to TFs, it is robust to annotation concerns (but - as the reviewer notes - limited to experimental data for the time being). In future studies, we hope to extend our methods by mapping GREs to TFs in a robust, data-driven way.

How many of the TFs binding logos they could recapitulate from the analysis if this info is not provided? What happens with TFs that don't change expression but are regulated by metabolites or posttranslational regulation? Out of the 40% TFs they could not find the binding sites, how many are non-dynamic?,

RESPONSE:

The reviewer highlights several concerns about GRE discovery. These are

important points. We note that GREs are discovered directly from genome sequence, so they are unaffected by TF dynamics or annotation.

(1) The number of TF binding logos discovered would not be affected if TFs were unknown. In our model, the “binding logos” (or GREs) are learned directly from genome sequences upstream of co-regulated genes (in biclusters). Consider *H. salinarum*, for example. Very few binding motifs for TFs are known in *H. salinarum* - nevertheless, we were able to discover 135 GREs. However, without knowledge of TF binding motifs (e.g., RegulonDB or from ChIP), it is very difficult to link the predicted GREs to their respective TFs, i.e. to know *which* TF binds these locations. As the referee notes, for dynamically changing TFs we may be able to perform the linkage using approaches similar to Inferelator. This, however, has some challenges because of the inability of these methods to distinguish direct from indirect TF influences. Nonetheless, this area of research is of great interest to us, and a subject of future study.

(2) The reviewer notes an important point that we did not emphasize sufficiently. EGRIN 2.0 can link TFs to GREs (and their predicted conditional activity) even when the TF doesn't change in expression. Other methods (including all “direct” methods included in DREAM5) rely on changes in TF mRNA levels to predict the genes they regulate. This may, in part, explain our model's increased performance on RegulonDB. We have added a sentence to the main text emphasising this point (Page 6, Lines 39-40, and Page 7, Lines 1-3).

How do the authors explain corems? ie groups of genes regulated the same way but by diff TFs? Are the diff TFs regulated the same way? could more general effects take place, like supercoiling, chromatin organization?

RESPONSE:

The reviewer highlights an interesting and important interpretation of corems. Since corems do not have to be regulated by a common factor, we have considered alternative explanations for their consistent co-expression (i.e. mechanisms; we note, however, that many corems are regulated by a common TF). This, in fact, is one of the key findings of our paper.

The reviewer notes several mechanisms that could achieve this. Like the reviewer, we also considered the possibility that multiple TFs controlling a corem could themselves be regulated in similar ways. In the *E. coli* PurR-ArgR example (Figure E30, Page 12, Line 38), we tried to understand this effect in detail. To our surprise, however, we observed low correlation between the expression levels of PurR and ArgR - even when the genes they controlled were highly correlated. Likewise, the TFs did not share common regulators (other than a common sigma factor). Instead, we noticed an interesting correlation between the measured levels of small molecules (PurR: guanine/hypoxanthine, and ArgR: arginine) that allosterically modulate the activity of these TFs. Thus, the data suggested that couplings between the activities of TFs (in addition to their expression levels), can cause the genes they regulate to have similar expression patterns. This is a dynamical interpretation of co-regulation, i.e. corems are the result of coupling of TF activities. This example is detailed in the main text (Page 12, Line 38).

The reviewer also notes that several general DNA structure-based mechanisms could lead to similar observations, including chromatin organization and supercoiling. This is an interesting idea that should be the subject of future study.

Since it is the observation that corems unite multiple regulatory mechanisms and subdivide others that we think distinguishes our work from the previous literature, we are encouraged that the reviewer understood this point and began considering specific mechanisms that could achieve it. We are excited to followup on this hypothesis in the future.

They also mentioned that many TF binding sites are not operational under certain conditions, how do they explain these cases in bacteria?

RESPONSE:

We thank the reviewer for noting a potential point of confusion in the manuscript. We do not claim that our model can predict when or if a particular GRE is physically bound by a TF. Rather, our model discovers which GREs are shared between genes when their expression is coordinated (and in what environments that occurs). This is an important distinction. Because we lack sufficient evidence for specific regulatory mechanism acting at each site, we avoid stating that GREs are physically bound by a TF in a given environment, since release of a TF could also accomplish coordinated down-regulation or de-repression. The GREs are, however, to be considered *operational* in the sense that the model predicts that they are important sites for co-regulation in the contexts in which they are discovered.

Biologically, the reviewer is likely well aware that binding sites can be “nonoperational” in bacteria for many reasons - many of which are directly tied to TF *activity*. For example, some allosterically modulated TFs can only bind GREs when their conformation is altered by a small molecule. TF expression can have similar consequences. Likewise, as mentioned in the reviewer’s previous comment, even DNA structure can influence whether or not sites are available for binding. All of these mechanisms can render particular GREs “non-operational” in some cases.

We reiterate, however, that the GRE discovery metric we define in the manuscript measures how often we discover a particular GRE upstream of a group of genes when they are co-regulated. Discovery in the context of coregulation suggests that these sites are functionally important for co-regulation, but it does not specify the particular physical mechanism (binding, unbinding, etc.).

The author say that their analysis will be extremely useful for synthetic biology, but i am doubtful. They find many nice correlations, they identify regulatory regions, but they cannot explain many of the observations (ie genes co-regulated although by didd TFs, binding sites not used etc...). In summary i think it is a nice story, but I would like a discussion of the limitations of the methodology.

RESPONSE:

We appreciate the reviewer’s concern for stating the limitations of the method. We did not wish to overstate the contributions of our model to synthetic biology. To address this concern, we have rewritten the discussion (Page 14: Lines 26-34 and Page 15: Lines 7-10) to clarify the limitations of our approach and suggest future directions.

That said, there are two features of our model that we think are valuable for synthetic biology: (1) our model can quickly infer comprehensive GRNs for understudied organisms, and (2) the model provides a quantitative framework for understanding the complicated relationship between regulatory DNA (GREs) and condition-specific coordination of gene expression.

It is taken for granted in synthetic biology that genes regulated by a common TF will be expressed similarly. Often, condition-specificity of regulation by a TF is ignored. We have shown, however, that this context matters. We have demonstrated that genes regulated by a common TF can have different coexpression patterns and respond differently to knockout of that TF. We have even shown that different TFs can regulate genes equivalently (at least in some environments). While we do not yet have a full mechanistic explanation for all of these cases, our model does allow us to quantitatively define these relationships. While the reviewer suggests that this will limit the usefulness of our model, we

think otherwise. We believe that a more nuanced understanding of gene regulation will allow investigators to design better circuits from endogenous components. For example, we can imagine designing condition-specific “switches” using native components (e.g., components described in the PurR/ArgR example exhibit such behavior; Figure 4). Implementing such designs could be as simple as taking endogenous promoters and attaching them to gene products one would like to co-regulate. Our model defines which parts can be used, in what combinations, and what their expected co-regulatory behavior will be. Perhaps more important, it can model the global context in which a designed circuit will operate. An investigator can predict which other processes are being co-regulated and thereby minimize interference with their circuit. In this sense, we believe that our model facilitates biologically-inspired engineering.

We have tried to clearly reflect both the limitations and opportunities of our method in a revised **Discussion** section.

We thank the referee for comments that have helped improve the manuscript.

Reviewer #3:

This paper combines information from genomic sequence and expression data to generate a map of co-regulated gene expression modules. In the paper the method is applied/validated to E.coli and H.Salinarium; applicability to other (sequenced) microbes makes this paper interesting for a broad audience.

RESPONSE:

We are grateful for the reviewer’s positive comments about our work - in particular recognizing that one of the most exciting aspects of our work is its potential for application to other sequenced microbes.

The authors show convincingly that the combination of several pre-existing methods for gene network reconstruction yields improved predictions of network organization, transcript isoforms, and promoter structure.

Validation of model predictions with experimental data and comparison with existing algorithms is performed adequately throughout all subsections.

The paper is clearly structured and well written, but in parts relies heavily on previous publications on EGRIN/cMonkey/inferelator making some sections difficult to understand for readers not too familiar with those papers.

RESPONSE:

Again, we thank the reviewer for noting the extensive validations performed to test the predictions of our model. We agree with the reviewer that this work builds on a series of three previously published papers (Reiss *et al.* 2006, Bonneau *et al.* 2007, and Bonneau *et al.* 2006) and can, therefore, be hard to follow if the reader is unfamiliar with these algorithms. We note, however, that these algorithms are widely known and used. They have even been the subject of comprehensive review (De Smet 2010).

In the spirit of limiting description of previously published material, we have addressed this concern in two ways: (1) We included a greatly expanded description of the algorithms in the Expanded View, and (2) Where relevant in the main text (**Materials and Methods:** Page 17, Lines 21-22 and **Introduction:** Page 5, Lines 3-5) we direct the reader to the **Expanded View** to learn more about the underlying algorithms.

Minor point: from the figure legends it remains unclear what the difference between data displayed in figures 3C and 4A is.

RESPONSE:

We thank the reviewer for highlighting a potential point of confusion in our figure legends. We have made the following specific changes to the legends to clearly indicate that **Figure 3C** refers to corems discovered in *H. salinarum* while **Figure 4A** refers to corems discovered in *E. coli*: (1) We modified the Figure legend titles for Figures 3-4 to indicate the organism described by the Figure: (Page 26, Line 16 and Page 27, Line 6) (2) We also added the species for each of the respective sub-legends (C and D - Page 26, Line 33 and Page 27, Line 10).

2nd Revision - authors' response

06 June 2014

Thank you again for submitting your work to Molecular Systems Biology. We have now heard back from the referee who was asked to evaluate your manuscript. As you will see below, this referee thinks that his/her main concerns have been satisfactorily addressed.

REFEREE REPORT

Reviewer #1:

The authors have addressed all my comments adequately.